# Confocal Laser Scanning Microscopy as a Method for Identifying Variation in Puparial Morphology and Establishing Characters for Taxonomic Determination

**DOI:** 10.3390/insects16010088

**Published:** 2025-01-16

**Authors:** Christian Elowsky, Leon Higley

**Affiliations:** School of Natural Resources, University of Nebraska-Lincoln, Lincoln, NE 68588, USA; celowsky@unl.edu

**Keywords:** Calliphoridae, blow fly, forensic entomology

## Abstract

Calliphoridae, or blow flies, are of much ecological and practical importance given their roles in decompositional ecology, medical and veterinary myiasis, and forensic entomology. As ephemeral and rapidly developing species, adults are frequently not present for identification, but puparia (the remaining outer integument of the third instar larvae) are frequently found. Historically, scanning electron microscopy (SEM) has been used for characterization, but it is not only time-consuming but also often expensive. Here, we demonstrate the use of confocal laser scanning laser microscopy (CSLM) for examining puparia. CSLM is significantly more rapid than SEM, requiring no preparation for imaging. For ten species of calliphorids, CSLM was used to image puparia. Not only did this provide characters for species identification but it also allowed for the examination of hundreds of specimens.

## 1. Introduction

The ability to identify and distinguish morphological features that are species-specific is a longstanding issue in the taxonomy of many groups, including the Insecta. This problem is compounded when the species or stage of interest has reduced morphological diversity, as is frequently the case with immature insects. These issues have great practical importance in forensic entomology because the identification of immatures to species is an essential activity [1].

Among one suborder of flies of great forensic importance is cyclorrhapous Diptera, where the immature stages that are the most challenging to identify are the pupae, or more exactly the puparium, which is the inflated and hardened exoskeleton of the third larval stage in which pupation occurs. Eclosed puparia (i.e., puparia from which the adult has emerged) are frequently found at death scenes with advanced decomposition and are sometimes associated with archeological remains that are hundreds to thousands of years old (e.g., [1,2,3]). Usually, these are puparia of blow flies, Calliphoridae, whose larvae feed on soft tissue during faunal decomposition.

Because the puparium is the exoskeleton of third-stage larva, the identification of puparia might seem to be identical to the identification of larvae. However, the formation of the puparium is associated with the contraction of the larva, collapse of the anterior segments, increased sclerotinization, and displaced or obscured of characters [4]. The larva exoskeleton is weakly sclerotized, so during the formation of the puparium, the larval exoskeleton is subject to changes in form, and with plates shifting dramatically as intersegmental membranes are stretched and hardened. Further obstacles to taxa identification exist in that larvae sizes are quite variable based on diet alone [5]. Also, in reduced resource situations, they may produce very light tan pupae [6] of reduced sizes, negating the utility of color and non-ratiometric landmarks on the puparium.

The problem is that practical keys specific to puparia do not exist for many important groups of cyclorrhapous dipterans, including the calliphorids. And to produce such keys, it is essential to identify stable, species-specific character stages [7].

### 1.1. Confocal Laser Scanning Microscopy

The use of confocal laser scanning microscopy (CLSM) is a recent development in the field of systematics and biology (e.g., [8,9]). While CLSM was developed in the 1950s, the first commercially available systems did not become available until the mid-1980s. Due to the prohibitive costs of these first systems, it was nearly two decades before organismal biologists began to develop techniques utilizing this technology.

CLSM utilizes the fluorescence properties of intrinsic biological molecules of the specimen or applied stains or labels. Simply a higher energy excitation light source, in this case a laser, illuminates the sample. If the physical chemistry of the receptor molecules allows absorption of these photons and emission at a lower energy (Stoke’s Shift), fluorescence occurs. These photons travel back through the system to a photomultiplier tube (PMT) having passed through a physical pin hole and several band pass filters. The pinhole results in a physically thin focal plane and the out-of-focus photons are discarded, unlike ubiquitous epifluorescence microscopes. This technique is thusly called “confocal”, literally “with focus”. A single image or scan is then a physically thin optical section of data from the sample. A series of images acquired with a motorized stage can then be acquired for increased visualization depth. Furthermore, this Z series can be merged through software to produce images or three-dimensional representations of the sample. CLSM is currently performed with between one and eight detection fluorophores per sample.

The development of CLSM systems has increased sensitivity with advances in optics and PMT technology. This has allowed for the use of intrinsic autofluorescence detection, which reduces artifacts from fixation and staining protocols, presumably. Furthermore, depending on the molecules being detected, photobleaching, which is signal degradation from the physical act of fluorescence, is an issue with applied fluorophores; autofluorescence is often far more stable through the imaging process.

Several areas of study in invertebrate morphology studies have utilized CLSM and intrinsic autofluorescence, including the study of genitalia, deposition of resilin, copepod, and mite morphology [10,11,12]. Zill et al. [10] utilized autofluorescence in cockroach legs in a study for biomechanical purposes. The legs were removed and cut longitudinally and imaged from both sides after mounting between two cover glasses. Images were acquired with only a single channel, which the authors concluded was a result of endogenous fluorescence from the cuticle. After image acquisition, combinations of the inner and outer image sets were produced for analysis. While this technique in developing research was well executed, the authors failed to completely understand basic biology and the physics of CLSM. By digesting away the muscles and dehydrating the cuticle, confirmation of the preservation of shape and morphology was not performed with other techniques. Furthermore, the cuticular waxes, which may have provided external morphology, were likely damaged; thus, the pigments obscured the autofluorescence and no other outer data were available.

In 2012, multiple channel acquisition was used for morphological investigations [11]. Mites which are quite small and nearly transparent were imaged externally for morphology and internally for genitalia. In this case, the external and internal features are autofluorescent in different wavelength channels. CSLM for mites is a boon as they are quite difficult to image via other means. In 2023, in another novel application super resolution, CSLM allowed for the nondestructive imaging of microfossils in amber [12].

In 2014, after nearly 15 years of publications, CSLM was finally used in the most natural method possible [13]. In this breakthrough study, the internal and diagnostic cephaloskeleton of larval Diptera was imaged. This structure is often dissected out or poorly imaged with compound microscopes, which results in artifacts of morphology or poor image quality as it is heavily sclerotized. The authors set the tone for the next decade of using autofluorescence and CSLM in adult and immature invertebrate imaging.

CSLM is not only faster than scanning electron microscopy but also superior to conventional light microscopy, especially when viewing structures known for fixation deformation and artifacting. The techniques used to image endogenous autofluorescence also involve no labeling and, in most cases, remove the dilemmas of photobleaching. If coupled with attention to refractive index mismatch issues, fixation and clearing protocols, and the limitations of the system, CSLM is a far more powerful tool than is currently utilized in morphological studies.

### 1.2. This Study

Given these needs to identify calliphorid puparia and opportunities offered by CSLM, our objectives in this study were as follows:

(1) To determine the usefulness of CSLM for identifying and characterizing puparia characters for species determinations (using forensically important calliphorid species as our model). In addition, provide comparative SEM elucidating the greater utility of CLSM for the identification of pupal characters.

(2) To examine the potential for using the automated generation of character states from CSLM imaging to provide data for species determinations.

(3) To determine how to allow evidence from CSLM to develop species identification tools for conventional light microscopy (thereby avoiding the problems of expense, limited access, and expertise associated with CSLM).

(4) To statistically evaluate morphological features for distinguishing various taxonomic levels through the use of multivariate analyses.

## 2. Materials and Methods

To allow simple, rapid examinations of puparia, no clearing or staining techniques were used. Puparia were mounted on standard insect pins in clay and imaged directly off the pins. Most puparia used were recovered directly after pupal eclosion and were mounted with the emerged fly. However, *Cochliomyia hominivorax* puparia were received in alcohol, so puparia were removed, allowing time for alcohol to evaporate, and then placed in a clay holder for imaging.

All imaging was conducted at the Morrison Microscopy Core Research Facility at the University of Nebraska-Lincoln. Species were examined, and source information is listed in Table 1. A total of 505 individual specimens were imaged, with imaging requiring approximately 15 min per specimen.

Comparisons of single samples per species were also made with a Nikon SMZ 800 dissecting microscope (Nikon Instruments Inc., Melville, NY, USA). Scanning electron microscopy was also performed. Pupal cases were desiccated in an oven for several days and then sputter-coated with gold–palladium and imaged on a Hitachi S-3000N Variable-Pressure SEM.

### 2.1. Insect Material

For the multivariate analysis, we examined the puparia of ten species within seven genera in three subfamilies of Calliphoridae (Table 1). For each species, a minimum of 25 puparia were imaged and measured, and where possible, additional individuals were included. All puparia examined were eclosed, and fully formed adults were recovered and then pinned with their puparia. By associating individual adults with the puparium from which they emerged, we could ensure that there were no errors in associating puparia with a given adult. Adult identifications were made using keys from Whitworth [14]) and Smith [15]. Independent confirmation of adult species was made by CGE, LGH, and Neal Haskell (Forensic Entomology Investigations).

The insect material used included field-collected (wild) and laboratory (cultured) fly populations. We recorded the location and host upon which the larvae had fed to examine the potential influence of these factors in confounding the identification of puparia.

### 2.2. Morphological Measurements

Examinations were conducted with a Nikon A1 Confocal Laser Scanning Microscope (CLSM). The CLSM was mounted on a Nikon Eclipse 90i automatic compound microscope, controlled with NIS-Elements 4.40 confocal acquisition software. A 4× objective was used for imagining puparia. A z-step of 16.4 μm was used, and image stacks were manually set between 30 and 85 images per stack to allow for a clear focus of the terminal end of the puparium. The self-contained calculation performed by NIS-Elements provides a 16.4 μm z-step based on the wavelengths used, physical properties of the optics, and to provide roughly an overlap of 1/3 of each image. This overlap allows for the best tradeoff between image numbers and a final maximum intensity projection image.

Images were produced with simultaneous, three-channel acquisition: channel 1 with excitation at 404.7 nm and emission at 425–475 nm, pseudocolored blue; channel 2 with excitation at 488 nm and emission at 500–550 nm, pseudocolored green; and channel 3 with excitation at 640.5 nm and emission at 663–738 nm, pseudocolored red. The final images were produced with NIS-Elements Software 1.0 and stored as TIF (tagged image format) files.

The initial evaluation of character states involved identifying characters that seemed to show intraspecific stability but interspecific variability in measurements made with ImageJ software. Figure 1 and Figure 2a,b illustrate the points measured and primary measurements. On each image, 12 points were identified by the user, specifying the beginning and end points for each of the six spiracular slits (Figure 1). Spiracular slits are a common character used in keys for calliphorid larvae and puparia. With this information, a macro was developed (Appendix A) and used in ImageJ, and this provided rapid determinations of distances and angles from these user-indicated points (Figure 2a,b).

### 2.3. Statistical Analysis

Data from ImageJ analyses were entered and summarized in Excel (Microsoft Office 2013). Data summaries include calculations of slit angles (from 90°) and slit lengths for slits in left and right spiracular plates. All angles were adjusted to a common, positive basis (angles from the right plate were subtracted from 180°) to allow for direct comparisons of angles from the two plates. Within plates, the widest portion was calculated as the distance from points with the narrowest and widest distances between slits; within and between plates were also calculated. Specifically, the widest and narrowest portions within each plate were indicated by the distance between points within a plate (widest: left plate 1–5, right plate 7–11; narrowest: left plate 2–6, right plate 8–12). Size, including both plate sizes, was indicated by the distance between points on different plates (widest 1–11; narrowest 6–8). Beyond these measures, other calculated variables included the total of the wide and narrow distances (with plates and across both plates) and the ratios of wide/narrow (again, within and across plates).

Because all *Cochliomyia hominivorax* were separated from other species by a single binary feature (partially closed spiracular labellum, Figure 3), we did not include *C. hominivorax* in the analyses. Similarly, because *Protophormia terraenovae* puparia could be distinguished by the presence of large papillae surrounding the spiracular area, it was also excluded from the analyses. For all analyses, we used SAS University Edition (SAS 9.4). We used stepwise canonical discriminate analysis to identify the most appropriate variables for discriminating taxonomic units. After this, we used canonical discriminate analysis for determining canonical variables and testing for significance. Our initial stepwise analyses included 1 characterization of perispiracular papillae (Figure 4 and Figure 5 show the degree of variation among species), 1 characterization of spiracular opercula (Figure 6, Figure 7 and Figure 8 show the degree of variation by subfamily for Luciliinae, Calliphorinae, and Chrysomyinae, respectively), 12 measurements from spiracular slits (6 lengths and 6 angles), 2 within-plate measures of width, 2 between-plate measures of width, 2 ratios of widths (wide/narrow) within and between plates, 4 means of plate width measurements, and 1 total of all slit lengths. Consequently, we considered 27 possible primary or secondary variables.

Among considerations in the final discriminate analysis were correlations between variables, normality, and differences in number of puparia by species. Regarding correlation, although our initial analysis included variables that were highly correlated with one another, our final 3 variables did not include such correlation. All variables used in our final testing show normal distribution of variation. Finally, although discriminate analysis does not depend on equal frequencies within group, we are cognizant that the strength of our results depends on representative samples from groups, so all analyses include males and females and different larval hosts and different geographic populations where possible (Table 1).

In addition to the use of discriminate analysis in our choice of discriminating variables, we also conducted analyses by subfamily, genus, and species to ensure that the variables used were consistent from broad to narrower taxonomic classifications. We also examined differences between sexes, location, and decompositional host where sufficient data were available with taxonomic groups.

## 3. Results

### 3.1. Limitations with Light Microscopy and SEM Versus CLSM

Light microscopy and SEM have limitations in characterizing calliphorid puparia. Figure 9 is a dissecting microscope image of a *Cochliomyia macellaria* puparium and demonstrates the conservative nature of the character of a puparium. Blow fly puparia are often darkly pigmented, translucent, and waxy, making light microscopy very difficult. Furthermore, underfed specimens might be quite light, and sizes vary greatly within a species, making most metrics far too unstable. SEM imaging, being based on surface structure data, negates many problems, but it is destructive, expensive, and time-consuming.

An advantage of CLSM imaging is the ability to look at multiple excitation parameters. The simultaneous recording of data from all channels occurred during z-step imaging; however, channel recordings could be reviewed separately in stored images. Sequential data collection was also performed for comparative analysis within the technique. Figure 10 demonstrates a four-channel sequential images series where 404.7, 488, 561, and 640.5 nm excitation is detected, respectively, into 425–475, 500–550, 575–625, and 663–738 nm emissions. Figure 11 is the same excitation and emission data but acquired simultaneously. In Figure 11, there is far more signal in the 500–550 and 575–625 channels; this is a result of either excitation or emission crosstalk. Figure 12, in comparison with Figure 10 and Figure 11, illustrates potential variation among individuals.

Crosstalk is often when a single excitation wavelength gives rise to multiple emission signals, and it is not possible to distinguish the source of an emission with a single excitation wavelength. Practitioners of confocal techniques are often faced with overlapping signals, making accurate colocalization difficult.

Figure 13 and Figure 14 present morphological features of taxonomic importance and their clarity with confocal maximum intensity projections.

Autofluorescence is quite low in the 500–500 nm and 575–625 nm channels with sequential acquisition (Figure 10). Sequential imaging acquisition resulted in the detection of a cuticular signal in 425–475 nm, which is probably the fluorescence of cuticular waxes and similar hydrocarbons based on the location of the signals on specimens. The chitin and scleritin matrix of the puparium wall were detected at 663–738 nm. Pseudocoloring only these two channels resulted in images which were either visually too complimentary or too contrasting for analysis. By pseudocoloring the 425–475 nm with both blue and green and merging those data with red pseudocolored 663–738 nm data, a serviceable image was produced. In this study, three-channel excitation and three-channel emission provided the most visually distinct images for analysis. Specifically, the laser wavelengths were channel 1 with excitation at 404.7 nm and emission at 425–475 nm, pseudocolored blue; channel 2 with excitation at 488 nm and emission at 500–550 nm, pseudocolored green; and channel 3 with excitation at 640.5 nm and emission at 663–738 nm, pseudocolored red (Figure 15). Crosstalk was not an issue in the analysis, however, because the 500–550 nm emission was not associated with pupal structures. On the contrary, crosstalk was helpful here because crosstalk emissions allowed for the differentiation of debris as debris rarely has the same dual channel emissions (Figure 16).

The similarity of the images from simultaneous and sequential imagining and the absence of detrimental issues from crosstalk (or more precisely, the benefit associated with crosstalk in this instance) allowed for simultaneous three-channel imaging for all subsequent examinations, with an obvious 67% reduction in the data acquisition. In this instance, 404.7 nm excitation led to emissions in the 425–475 nm range but also to the 500–550 nm range, which were ordinarily associated with excitation by the 488 nm laser.

Figure 9, Figure 13 and Figure 14 clearly demonstrate the clarity of characters offered by CSLM over conventional nondestructive light microscopy. Figure 14 provides clear morphological features and landmarks as well, and while it is a maximum intensity projection, the data show a depth character, making the hinge quite apparent as well. Figure 8 shows two images of two puparia from *Cochliomyia macellaria* which did yield adult flies. The difference was that the upper puparium was darkly pigmented and the lower was light tan and translucent. Both resulted in datasets sufficient for analysis. This study did not investigate the amounts of hydrocarbons produced, but the lower image is significantly brighter in the 425–475 nm and 500–550 nm channels at the same acquisition settings.

The use of a 4× objective allows for the visualization of minute structures if they are covered in cuticular hydrocarbons. Figure 13 highlights setae which are roughly 10–15 μm in total size. At similar total magnifications via SEM and light microscopy, these features are indiscernible (Figure 17A). At slightly higher magnifications, either digitally or optically (Figure 16), ultrafine details may be made out. 

### 3.2. Image Comparisons of Calliphorid Species by CLSM and SEM

Figure 18, Figure 19, Figure 20, Figure 21, Figure 22, Figure 23, Figure 24, Figure 25, Figure 26, Figure 27, Figure 28, Figure 29, Figure 30, Figure 31, Figure 32, Figure 33, Figure 34, Figure 35, Figure 36 and Figure 37 are dual images of 10 species comparing CSLM and SEM at 40×.

### 3.3. Results of Statistical Analysis

Tests of all 27 variables from the stepwise canonical analysis provided models with significant F values for inclusion of 21 of the 27 variables. However, consideration of individual contributions to the total model r^2^ indicated that only three variables were necessary to produce an average r^2^ in excess of 0.85: for species, unweighted r^2^ = 0.83, r^2^ weighted by variance = 0.81; for genera, unweighted r^2^ = 0.81, r^2^ weighted by variance = 0.79; and for subfamilies, unweighted r^2^ = 0.75, r^2^ weighted by variance = 0.71.

Some significant differences in values were noted across and within species for many measured and calculated variables for sex, location, and diet. However, none of these differences were reflected in subsequent discriminant analyses; the same canonical variates emerged when analyses were conducted by sex, location, and diet, as were found across sex, location, and diet.

Descriptive statistics for subfamily are presented in Table 2. Results for the stepwise canonical discriminate analysis are presented in Table 3, with all variables significant by F test at *p* < 0.0001. Results for the discriminate analysis are present in Table 4, Table 5 and Table 6, and again all variables were significant by F test at *p* < 0.0001, as was the model itself by Wilks’ Lambda and other tests. A scatter plot indicating separation of species by canonical variates 1 and 2 is presented in Figure 38.

Descriptive statistics for genera are presented in Table 7. Results for the stepwise canonical discriminate analysis are presented in Table 8, with all variables significant by F test at *p* < 0.0001. Results for the discriminate analysis are present in Table 9, Table 10 and Table 11, and again all variables were significant by F test at *p* < 0.0001, as was the model itself by Wilks’ Lambda and other tests. A scatter plot indicating separation of genera by canonical variates 1 and 2 is presented in Figure 39.

Finally, descriptive statistics for ten calliphorid species are presented in Table 12. The results for the stepwise canonical discriminate analysis are presented in Table 13, with all variables significant by F test at *p* < 0.0001. Results for the discriminate analysis are present in Table 14, Table 15 and Table 16, and again all variables were significant by F test at *p* < 0.0001, as was the model itself by Wilks’ Lambda and other tests. A scatter plot indicating separation of species by canonical variates 1 and 2 is presented in Figure 40. As these results indicate, the differences between genera are more discernable than between species within genera.

## 4. Discussion

Historically, images of immatures and pupal stages of blow flies have been taken with scanning electron microscopy. This technique is inferior for several reasons. First, it does not allow for more than only surface comparisons of materials being viewed, and due to the method, all materials appear the same. Secondly, SEM imaging often requires complete dehydration, metal sputter coating, and the sample to be placed in a vacuum. This is a slow and destructive process which is not appropriate for anthropology, archeology, or forensic entomological applications. While SEM does allow for very high magnifications, it is of limited utility for identification. Finally, the dehydration of larvae, before the pupal stage, causes severe artifacting and wrinkling.

Confocal analysis of the spiracular region provides a richer morphological character set than previous methods. Figure 9 at 1000× demonstrates dentate setae, which have not been reported before in the literature. These features correspond with the areas in all species below the spiracular slits. Furthermore, when feeding all the maggot species, open and closed spiracular labellum is observed, thereby not only covering the spiracular plates but presumably using the dentate setae to clean the slits. This basic observation resulted from the observations of the hydrocarbon tipped setae.

As confocal imaging causes no measurable damage to a specimen and entire specimens may be imaged, there is no destruction involved. This lack of destruction maintains context, including in forensic applications. Furthermore, if only the spiracular regions are imaged, accounting for mounting and acquisition, dozens of samples may be imaged in a few hours. This makes CSLM far superior to any other method utilized in the past or currently.

The premise that CSLM can provide insights into the identification of morphological characters and character states for species and other taxonomic divisions is well supported by the results here. The increased contrast and pseudocolor representation of features (such as perispiracular papillae, e.g., Figure 4 and Figure 5) provides a basis for recognizing the variation in features that can be difficult to determine with conventional light microscopy.

If CSLM were less expensive, required less user expertise, and was widely available, we can imagine that it would become the preferred tool for immature insect identification. Currently, we think that CSLM is better used as a research tool for identifying appropriate characters and character variation for use with conventional light microscopy. The characters that emerged in this study for identifying puparia to subfamily, genera, and species (Table 2, Table 3, Table 4, Table 5, Table 6, Table 7, Table 8, Table 9, Table 10, Table 11, Table 12, Table 13, Table 14 and Table 15) can all be seen and characterized with light microscopy without dissections and or sample destruction.

This approach requires image analysis (at 40×), but free software (e.g., ImageJ) is available for such analyses and offers the additional advantage of providing a visual and mathematical record of data used in species determination. Moreover, because of automated calculations, in tests with conventional light microscopes, we found that we could go from pinned samples to output data as quickly as 50 specimens per hour (a huge improvement over time needed for conventional identification).

Given the potential importance of identifying calliphorid puparia to species for archeological, anthropological, and forensic purposes, it is strange that keys for puparia identification do not exist. Undoubtedly, the misconception that larval characters are identical in puparia (as the puparium is the modified exoskeleton of the third-stage larva) must be associated with this omission. However, our results, with a very stable morphological region, spiracle plates, and associated tissues, show substantial inter- and intraspecific variation.

Our results demonstrate that spiracular structures, especially spiracular slits and plates, offer sufficient stability and variation to provide a basis for identification. However, in developing a scheme for using these characters, we had to face some myths that underlie much species identification. Dichotomous keys imply that a series of binary choices are possible for arriving at an identification, and an explicit treatment of variation is typically missing. In our examination of these ten species, we found only two characters, the folded spiracular labellum (exclusive to *Cochliomyia hominivorax*) and the large spiracular papillae (exclusive to *Protophormia terraenovae*), that were truly dichotomous. Instead, a range of stages and characters in combination were necessary for reliable identification (and, yes, having made this statement, we do appreciate the irony in our having to use dichotomous keys for adult flies in setting our species categories in this study).

A related mythological issue is the degree to which character states are representative of a species, stage, or population. In taxonomic revisions, the breadth of geographic variation and number of specimens examined are commonly reported. In contrast, we found no reports on background sources or number of specimens examined for widely used keys to calliphorid larvae. We explicitly considered variation by sex, location, and diet (Table 1), and not surprisingly, we found a variation in our ability to identify material to species (Figure 19).

The variables and character states we identified through discriminate analysis showed taxonomic consistency from subfamily to genus to species; the same variables emerged in the same order of importance from stepwise canonical discriminate analysis. This supports the view that these are genuinely species-specific discriminators. However, variation exists within these traits.

The most striking evidence of variation occurred with *Phormia regina* (Figure 18 and Figure 19). As scatter plots of canonical variates 1 and 2 illustrate, of the 106 *P. regina* examined, 12 clustered with *Calliphora livida.* With closer examination, this variation is not explained by sex, location, or diet: the 12 individuals occur in various populations, both sexes, and with various diets. Consequently, the source of this variation is beyond our ability to resolve with our data.

The example of variation in *P. regina* leads to an important question: how many individuals are necessary to determine a species identification? Short of DNA sequencing or a similar definitive test, identification necessarily includes a degree of uncertainty. This uncertainty is rarely stated or quantified, and it is one of the hallmarks of true experts on a taxon that they have a gestalt or intuitive sense of the range of variation in a species. Dichotomous keys can falsely imply a degree of certainty that ignores genuine variation.

The approach used here, with species characteristics indicated through discriminate analysis, offers one answer to the numbers of specimens for identification question. While *C. hominivorax* and *Protophormia terraenovae* can be identified with 100% certainty (based on the folded spiracular labellum and large spiracular papillae, respectively), based on the 12 of 106 misclassification rate, an individual *P. regina* identification using our method has an 11.3% chance of being wrong. By simple probability, identifying four individuals would reduce the chance of misclassifying a group of *P. regina* to only 0.016%. For both scientific and forensic purposes, the ability to place a probability on identification offers great advantages, but this is only possible if the underlying error rate in an identification procedure is made explicit.

## Figures and Tables

**Figure 1 insects-16-00088-f001:**
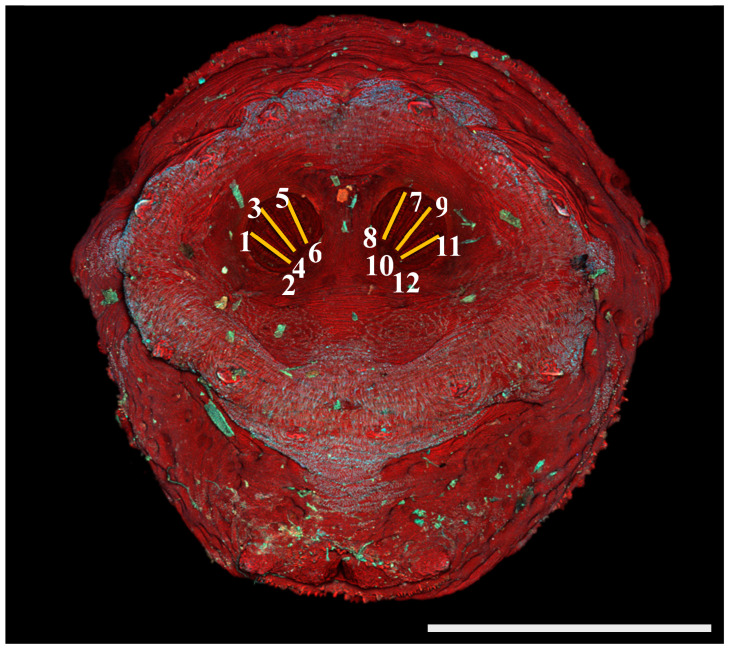
*Phormia regina* puparium. Confocal maximum intensity projection at 40×, scale bar is 1 mm. Coordinates for start and stop points (points 1–12) on spiracular slits are reported and used for morphometric calculations.

**Figure 2 insects-16-00088-f002:**
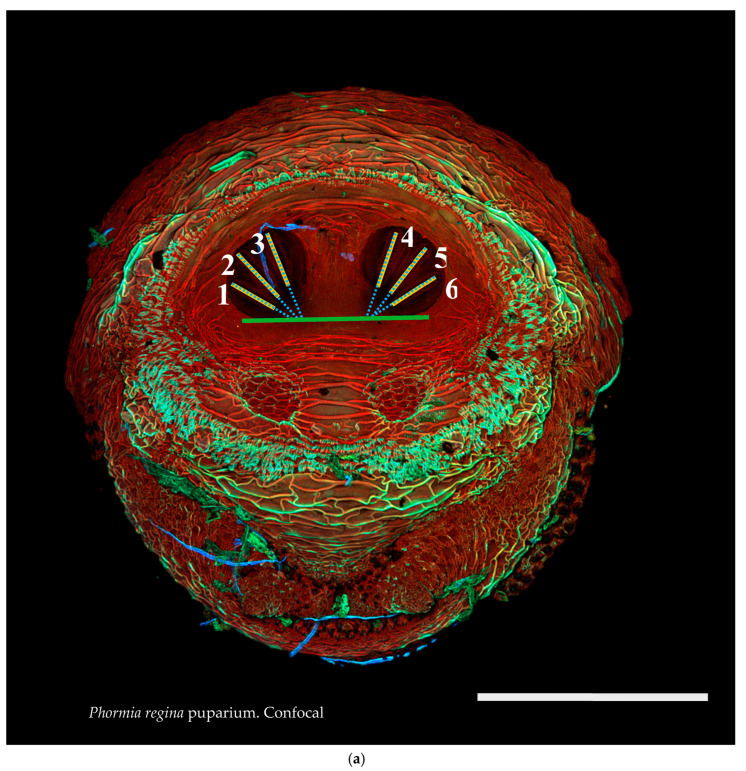
(**a**) *Cochliomyia macellaria* puparium. Confocal maximum intensity projection at 40×, scale bar is 1 mm. Green line is a representation of the 180-degree line set by the ImageJ 1.0 macro from user input. Quantification of angles (blue dashed lines) and length of spiracular slits from position 1 to 6. (**b**) Diagram of measurements and calculations. Dotted gray line is user-set horizontal or 180°; dotted black lines indicate calculated angles. Thick black lines are length measurements of spiracular slit start and stop points. Horizontal black line is calculated distances based on start and stop points reported by ImageJ. Black triangles are papillae. SO: spiracular operculum (scored as easily viewed [1>] or not [0]). P: papillae (scored as not easily visible [0], visible [1], or large [2]).

**Figure 3 insects-16-00088-f003:**
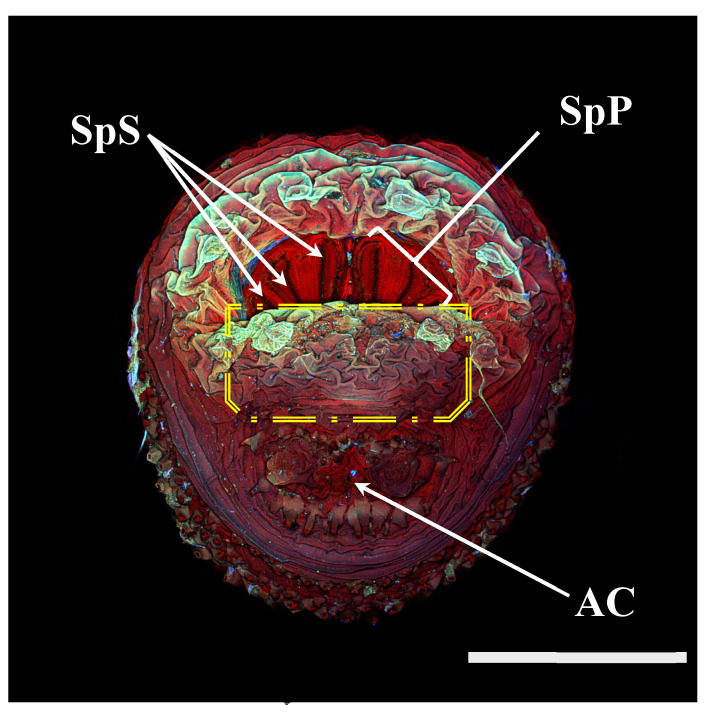
*Cochliomyia hominivorax* puparium. Confocal maximum intensity projection at 40×, scale bar is 1 mm. Yellow polygon highlights spiracular labellum which obscures spiracular plates and spiracular opercula. SpS: spiracular slits; SpP: spiracular plate; AC: anal cleft.

**Figure 4 insects-16-00088-f004:**
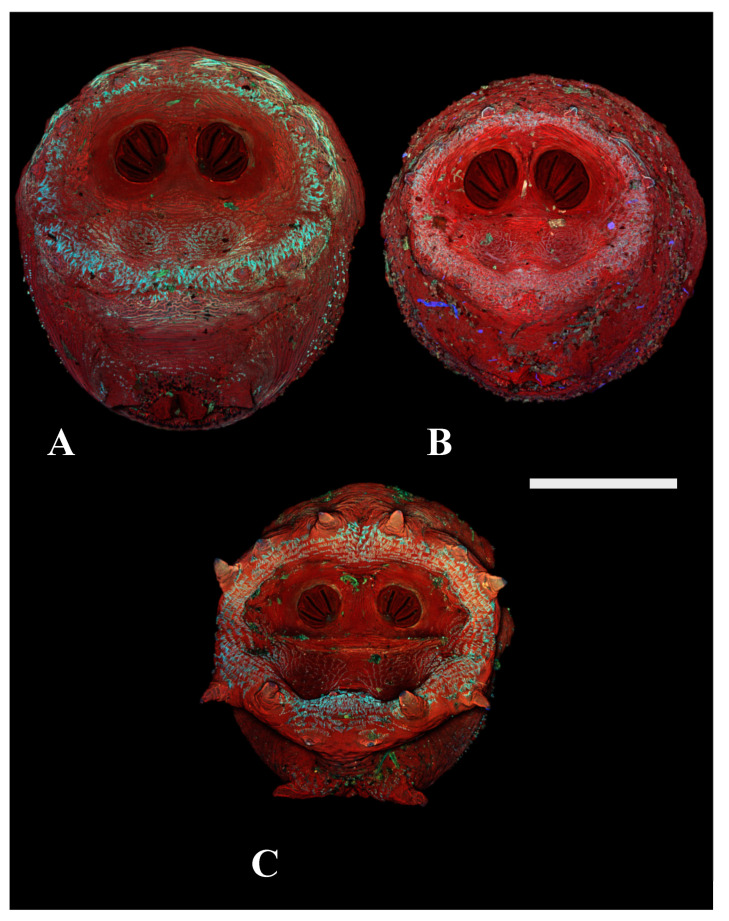
Confocal maximum intensity projections of puparia at 40×, scale bar is 1 mm. (**A**) *Chrysomya megacephala* scored as a “0” for lacking prominent papillae. (**B**) *Phormia regina* scored as a “1” for presence of perispiracular papillae. (**C**) *Protophormia terraenovae* scored as a “2” with very prominent perispiracular papillae, in this case visible with the naked eye.

**Figure 5 insects-16-00088-f005:**
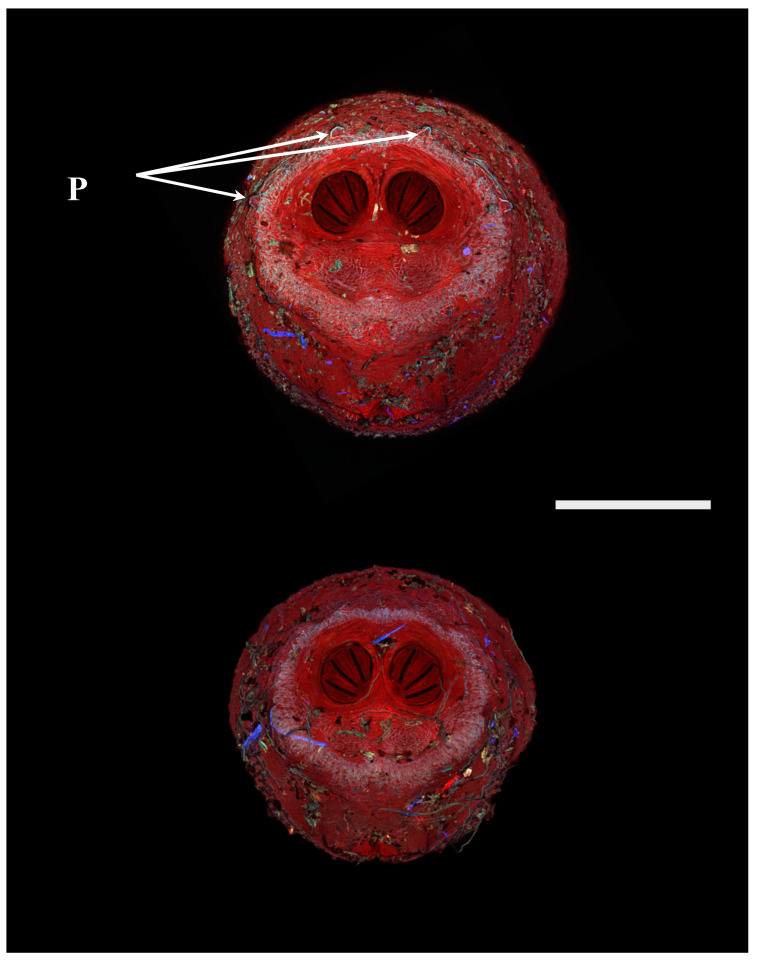
*Phormia regina* puparia. Confocal maximum intensity projection at 40×, scale bar is 1 mm. Upper was scored as “1” for visible papillae, lower was scored at “0”. P: perispiracular papillae.

**Figure 6 insects-16-00088-f006:**
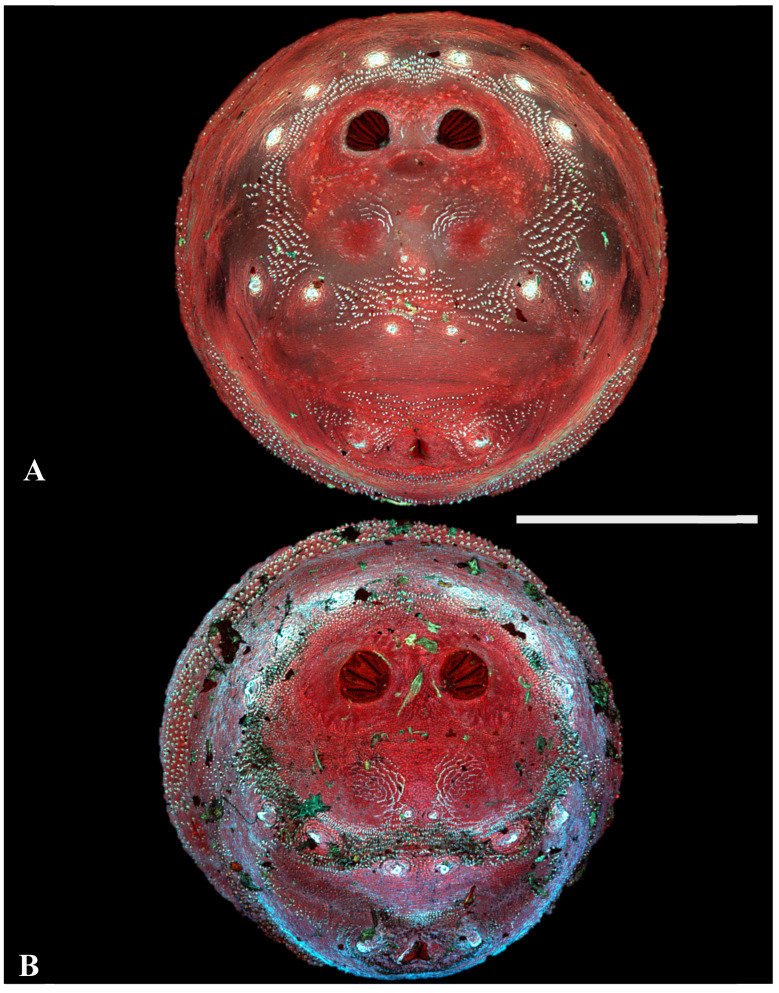
Luciliinae. Confocal maximum intensity projections of puparia at 40×, scale bar is 1 mm. (**A**) *Lucilia sericata;* (**B**) *L. coeruleiviridis*.

**Figure 7 insects-16-00088-f007:**
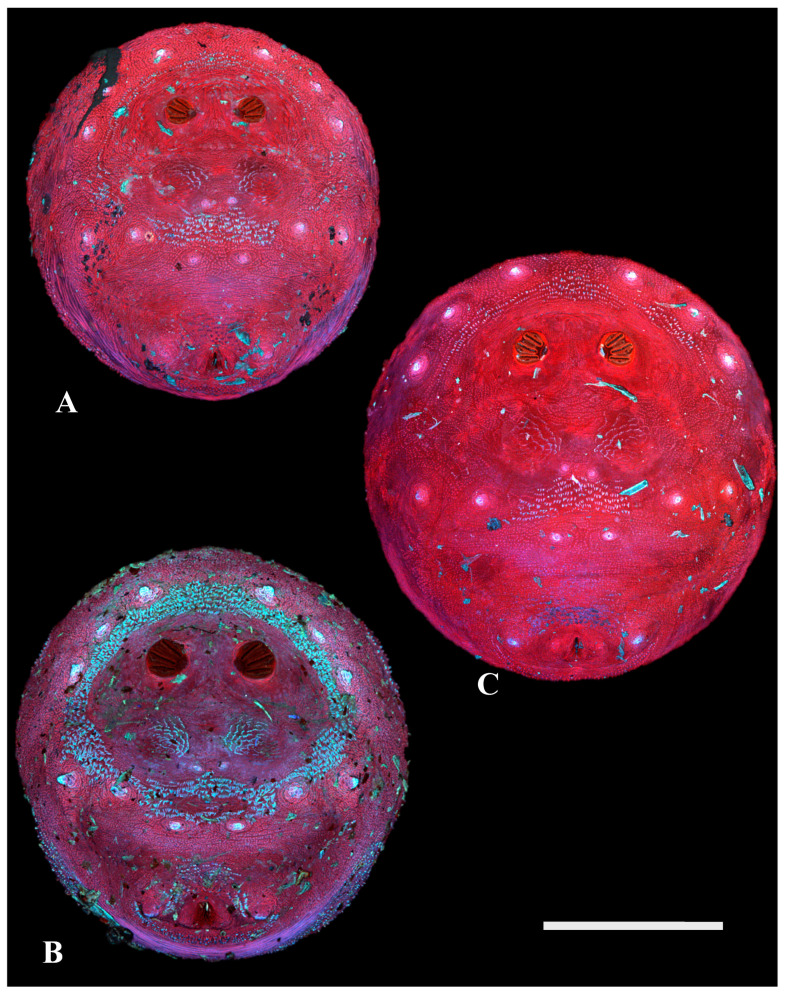
Calliphorinae. Confocal maximum intensity projections of puparia at 40×, scale bar is 1 mm. (**A**) *Calliphora livida,* (**B**) *Calliphora vicina,* and (**C**) *Cynomya cadaverina*.

**Figure 8 insects-16-00088-f008:**
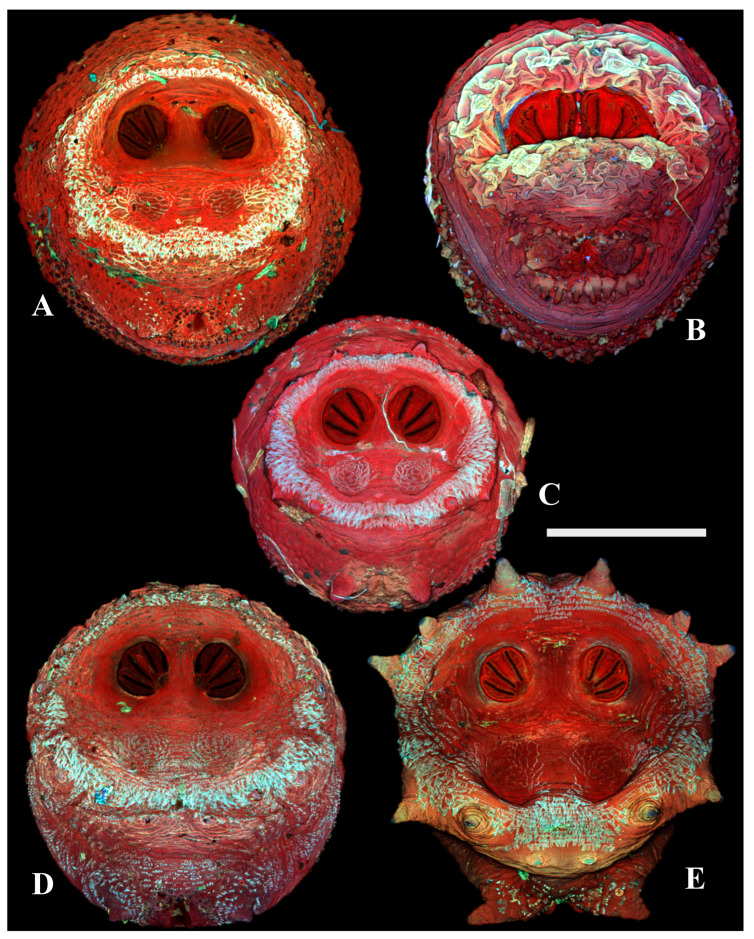
Chrysomyinae. Confocal maximum intensity projections of puparia at 40×, scale bar is 1 mm. (**A**) *Cochliomyia macellaria*, (**B**) *Cochliomyia hominivorax,* (**C**) *Phormia regina,* (**D**) *Chrysomya megacephala,* and (**E**) *Protophormia terraenovae*.

**Figure 9 insects-16-00088-f009:**
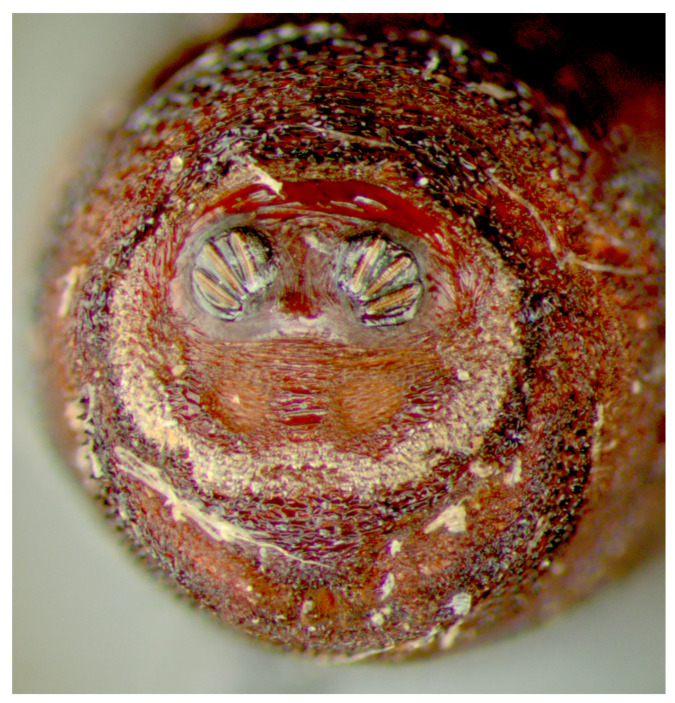
*Cochliomyia macellaria* puparium, dissecting microscope. Dissecting microscope image at a total magnification of 40×. Spiracular plates with slits and spiracular opercula easily viewed at this minimal magnification. Note pigmented perianal spines (dentate setae), which appear dark.

**Figure 10 insects-16-00088-f010:**
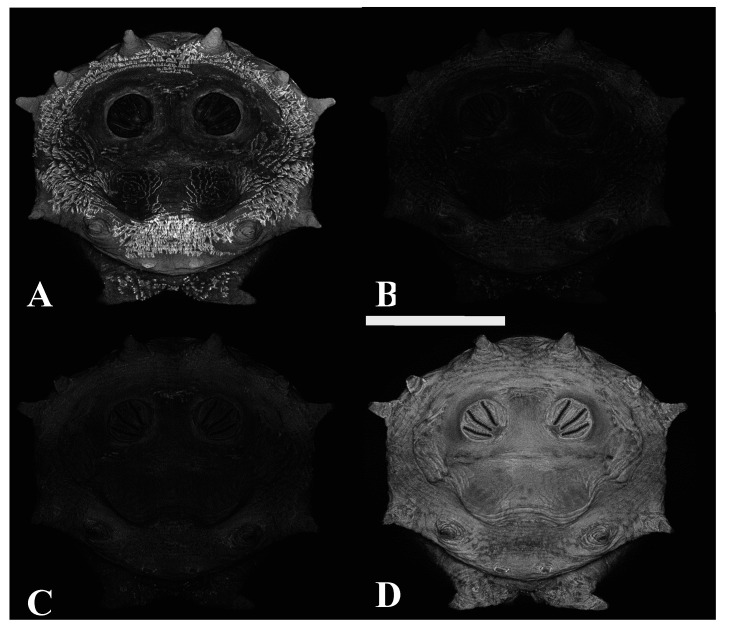
*Protophormia terraenovae* puparium. Imaged sequentially with four excitation lasers with four emissions at 40×, scale bar is 1 mm. Data shown are a maximum intensity projection of all images. (**A**) The 404.7 nm excitation and data are 425–475 nm emission. (**B**) The 488 nm excitation and data are 500–550 nm emission. (**C**) The 561 nm excitation and data are 575–625 nm emission. (**D**) The 640.5 nm excitation and data are 663–738 nm emission. Autofluorescence is quite low in the 500–500 nm and 575–625 nm channels with sequential acquisition.

**Figure 11 insects-16-00088-f011:**
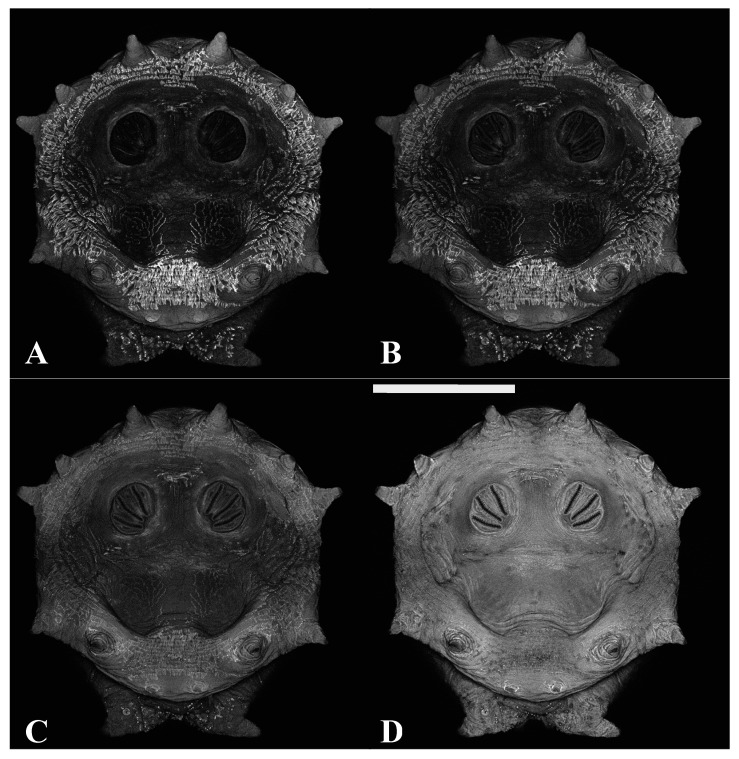
*Protophormia terraenovae* puparium. Imaged with four excitation lasers with four emissions simultaneously at 40×, scale bar is 1 mm. Data shown are a maximum intensity projection of all images. (**A**) The 404.7 nm excitation and data are 425–475 nm emission. (**B**) The 488 nm excitation and data are 500–550 nm emission. (**C**) The 561 nm excitation and data are 575–625 nm emission. (**D**) The 640.5 nm excitation and data are 663–738 nm emission. (**C**) Contains similar data as three other channels, providing no contrast between signal areas.

**Figure 12 insects-16-00088-f012:**
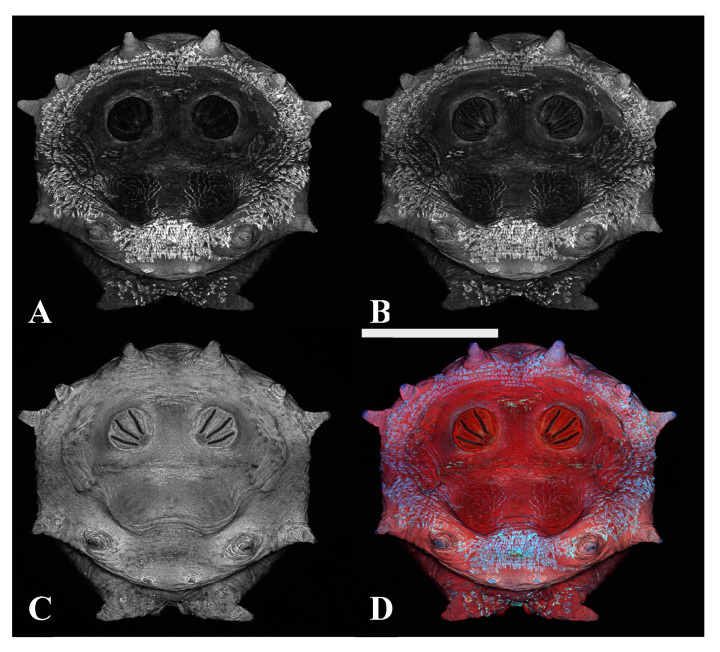
*Protophormia terraenovae* puparium. Imaged with three excitation lasers with three emissions simultaneously at 40×, scale bar is 1 mm. Data shown are a maximum intensity projection of all images. (**A**) The 404.7 nm excitation and 425–475 nm emission. (**B**) The 488 nm excitation and 500–550 nm emission. (**C**) The 640.5 nm excitation and 663–738 nm emission. (**D**) Merging of three datasets: A = blue, B = green, C = red.

**Figure 13 insects-16-00088-f013:**
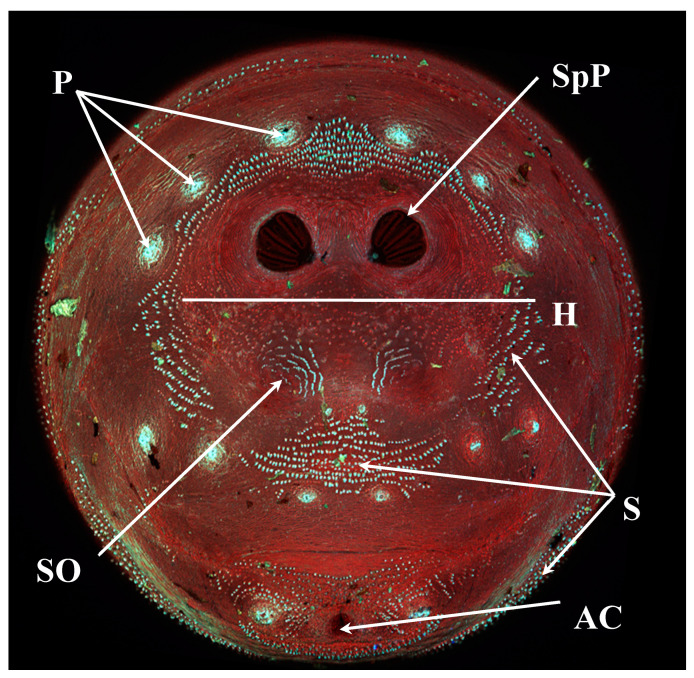
*Lucilia sericata* puparium. P: papillae; SpP: spiracular plate; H: hinge; S: setae; SO: spiracular operculum; AC: anal cleft.

**Figure 14 insects-16-00088-f014:**
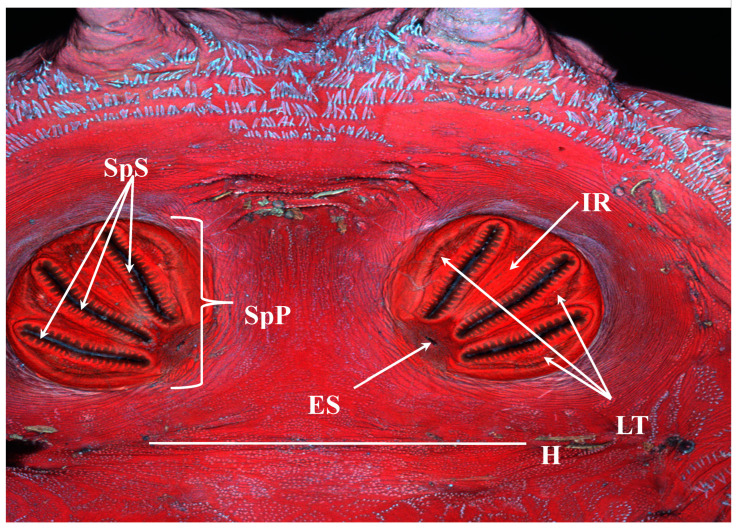
*Protophormia terraenovae* puparium. Confocal maximum intensity projection. SpS: spiracular slits; SpP: spiracular plate; ES: ecdysial scar; LT: lateral thickenings; IR: intermediate ray; H: hinge.

**Figure 15 insects-16-00088-f015:**
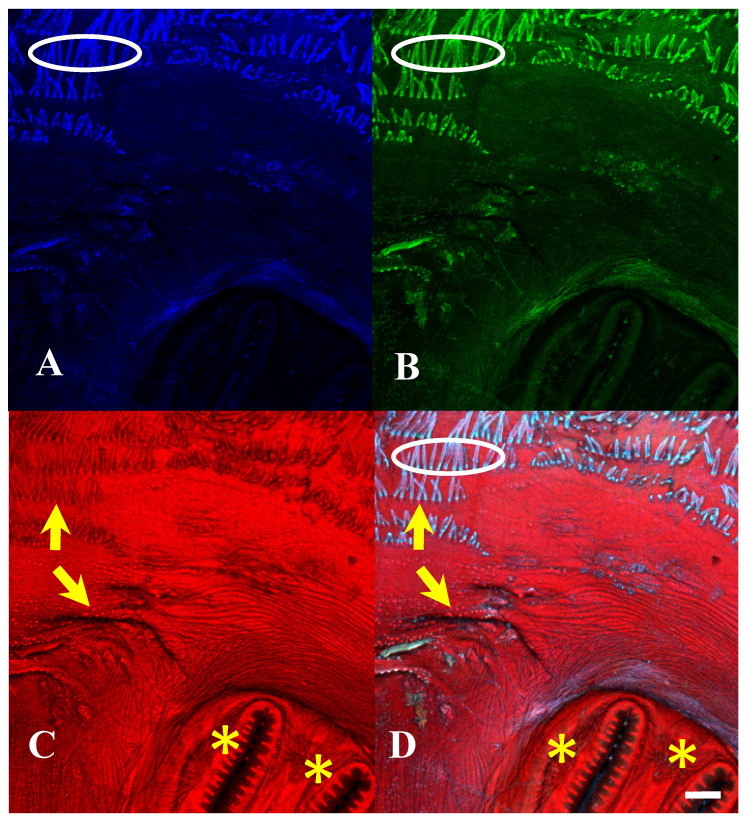
*Protophormia terraenovae* puparium. Imaged with three excitation lasers with three emissions simultaneously at 100×, scale bar is 500 µm. (**A**) The 404.7 nm excitation and data are 425–475 nm emission. (**B**) The 488 nm excitation and data are 500–550 nm emission. (**C**) The 640.5 nm excitation and data are 663–738 nm emission. (**D**) Merge of all three datasets: (**A**) = blue, (**B**) = green, (**C**) = red. Yellow arrows in (**C**,**D**) indicate scleriterized structures; yellow * indicates “spiracular hair cluster”, which is not fluorescent but is slightly visible due to refraction of the signal from the other autofluorescence signals, sometimes found on slits and rays. White ovals in (**A**,**B**,**D**) indicate bright autofluorescence from hydrocarbons on setae.

**Figure 16 insects-16-00088-f016:**
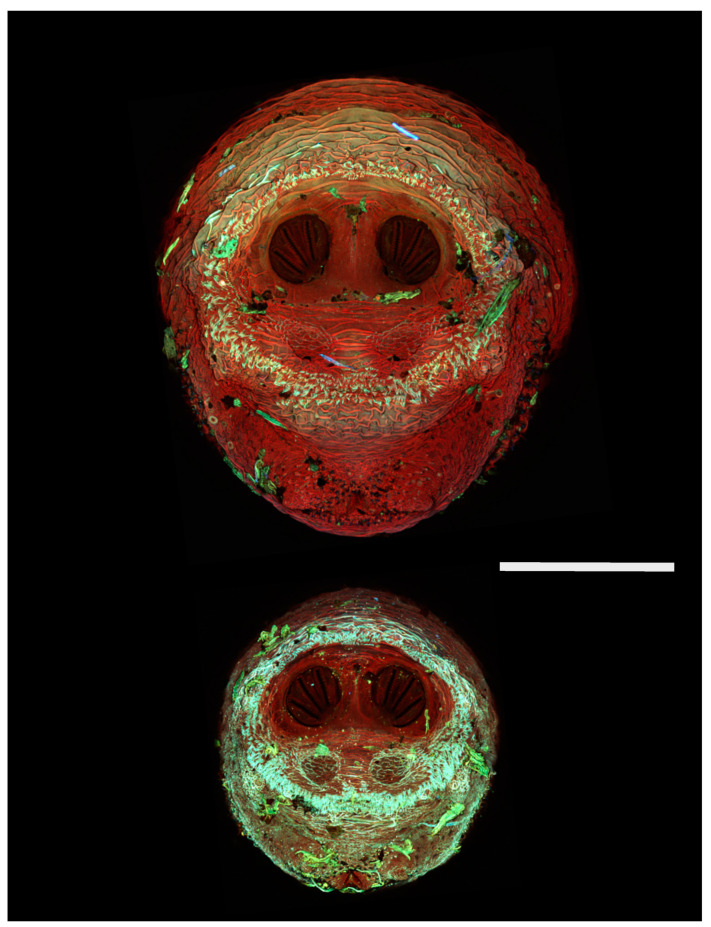
*Cochliomyia macellaria* puparium. Confocal maximum intensity projection at 40×, scale bar is 1 mm. Both samples came from the same cohort; the lower was an extremely lightly colored puparium, the upper more typical. Note that the spiracular plate and slit sizes are nearly identical. Blue and light green debris is easily differentiated from puparium.

**Figure 17 insects-16-00088-f017:**
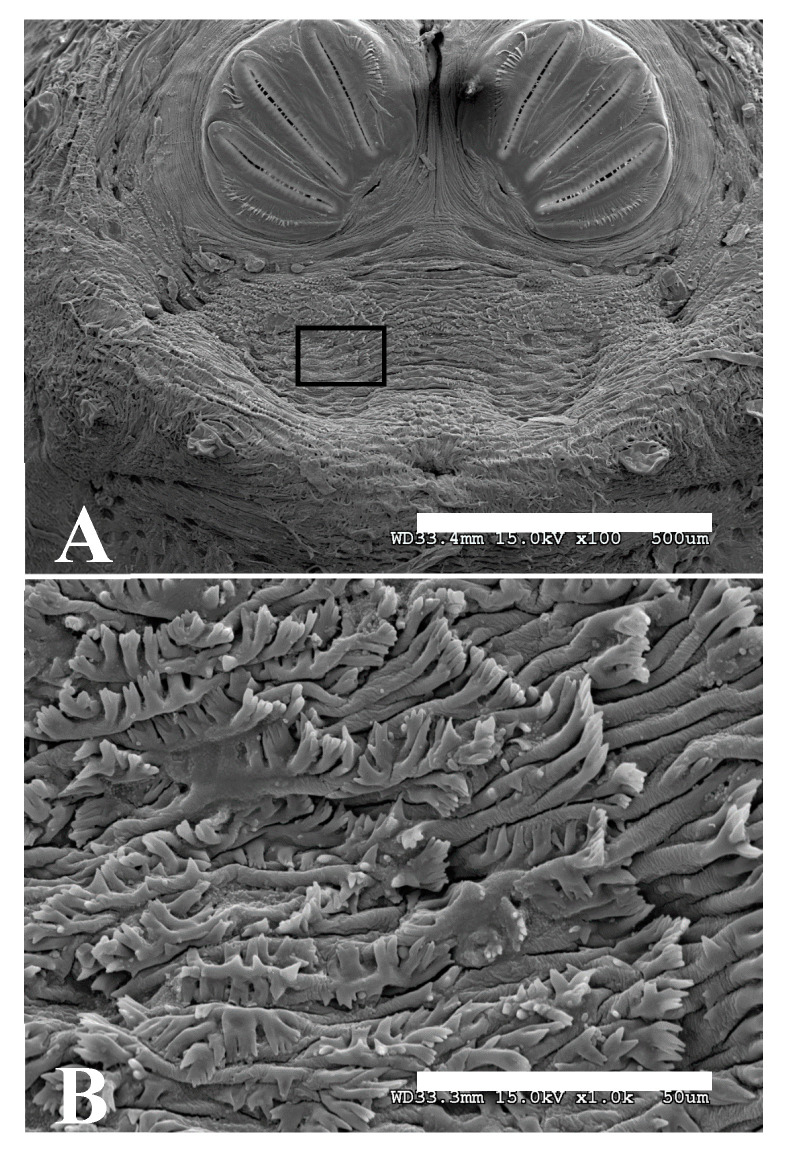
*Phormia regina* puparium. (**A**) Scanning electron micrograph at 100×, scale bar is 500 µm. Black box is in a spiracular operculum and used as insert for (**B**). (**B**) Scanning electron micrograph at 1000× scale bar is 50 µm. Tips of setae which are covered in hydrocarbons are comb-like in appearance.

**Figure 18 insects-16-00088-f018:**
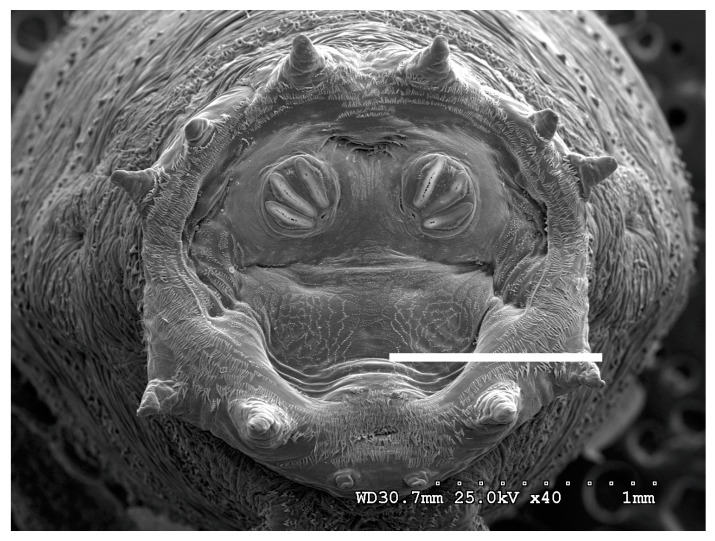
*Protophormia terraenovae* puparium. Scanning electron micrograph at 40×, scale bar is 1 mm. Spiracular plates with slits are visible at this minimal magnification. Ecdysial scars are also visible. Prominent papillae are present in this species. Spiracular opercula are quite smooth but visible. “Spiracular hair cluster” visible bordering the spiracular slits. Intermediate rays are weakly visible here.

**Figure 19 insects-16-00088-f019:**
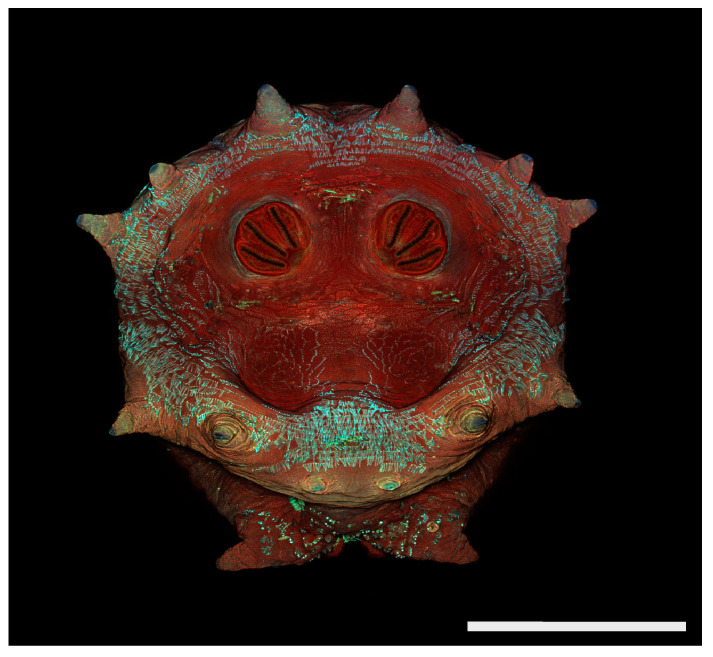
*Protophormia terraenovae* puparium. Confocal maximum intensity projection at 40×, scale bar is 1 mm. Spiracular plates with slits and spiracular opercula are easily viewed at this minimal magnification due to contrasting coloration of structures. Intermediate rays are weakly visible here.

**Figure 20 insects-16-00088-f020:**
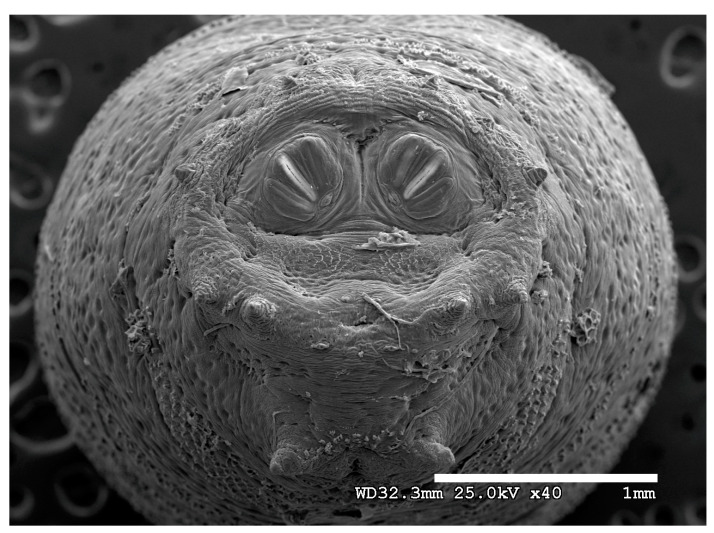
*Phormia regina* puparium. Scanning electron micrograph at 40×, scale bar is 1 mm. Spiracular plates with slits and sunken and deeply sculptured spiracular opercula are easily viewed at this minimal magnification. Ecdysial scars are also visible. Papillae present.

**Figure 21 insects-16-00088-f021:**
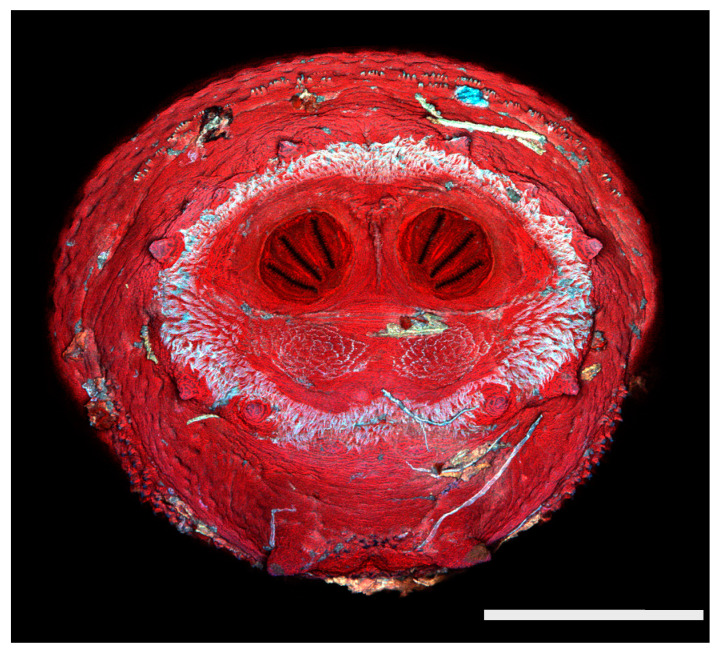
*Phormia regina* puparium. Confocal maximum intensity projection at 40×, scale bar is 1 mm. Spiracular plates with slits and spiracular opercula are easily viewed at this minimal magnification.

**Figure 22 insects-16-00088-f022:**
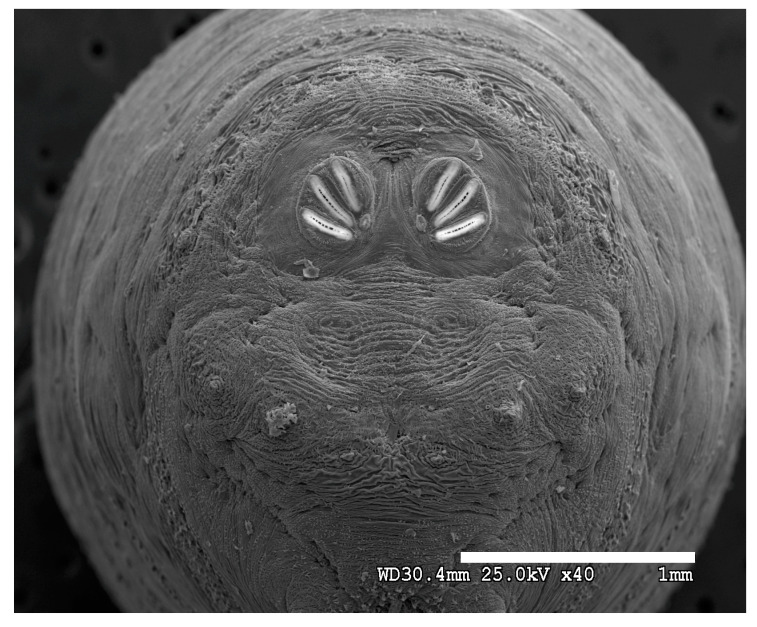
*Chrysomya megacephala* puparium. Scanning electron micrograph at 40×, scale bar is 1 mm. Spiracular plates with slits and spiracular opercula are easily viewed at this minimal magnification. Ecdysial scars are also visible.

**Figure 23 insects-16-00088-f023:**
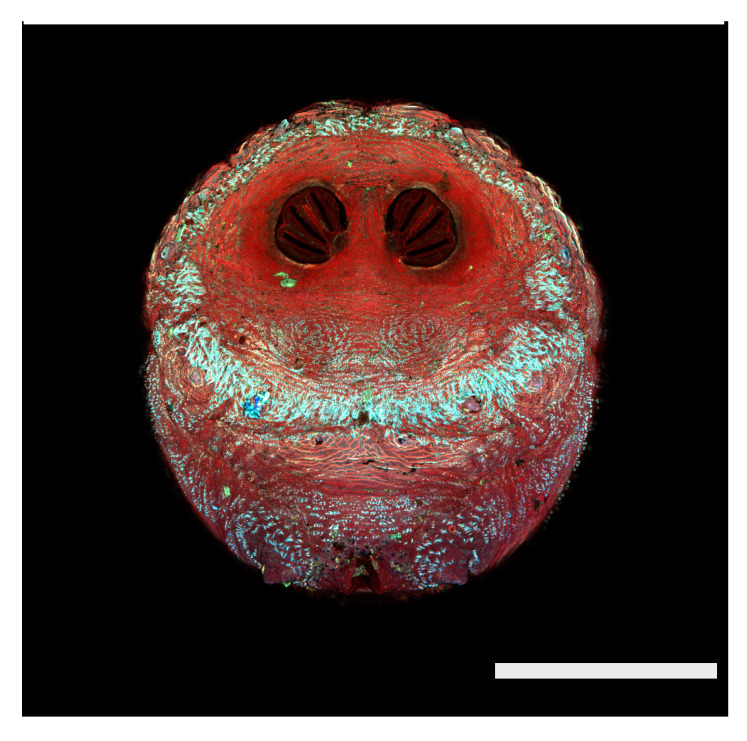
*Chrysomya megacephala* puparium. Confocal maximum intensity projection at 40×, scale bar is 1 mm. Spiracular plates with slits and spiracular opercula are easily viewed at this minimal magnification. Ecdysial scars are also visible.

**Figure 24 insects-16-00088-f024:**
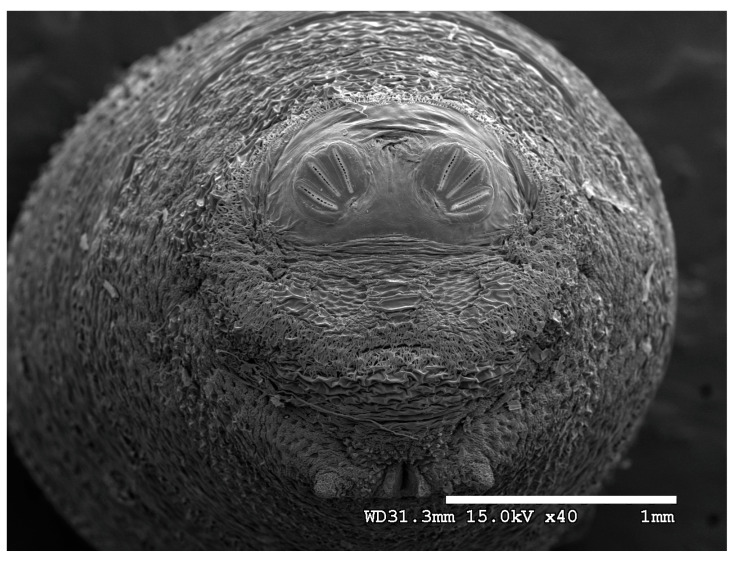
*Cochliomyia macellaria* puparium. Scanning electron micrograph at 40×, scale bar is 1 mm. Spiracular plates with slits and spiracular opercula are easily viewed at this minimal magnification. Ecdysial scars are also visible.

**Figure 25 insects-16-00088-f025:**
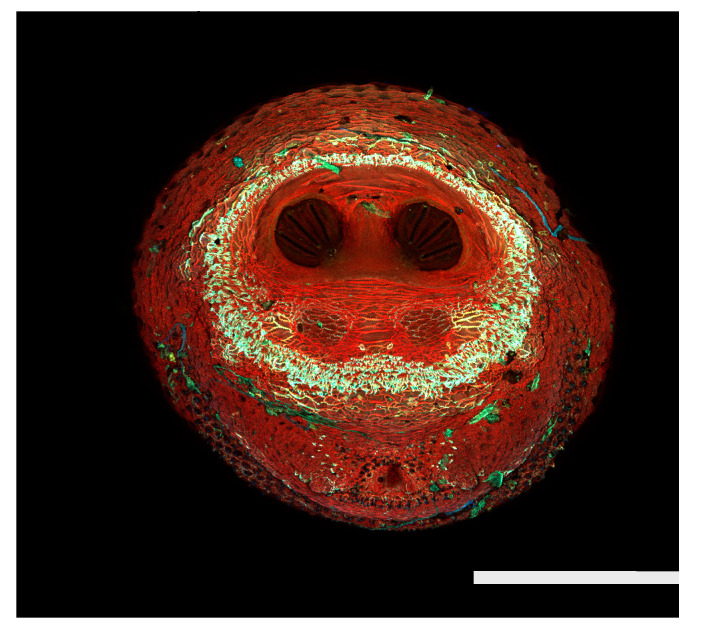
*Cochliomyia macellaria* puparium. Confocal maximum intensity projection at 40×, scale bar is 1 mm. Spiracular plates with slits and spiracular opercula are easily viewed at this minimal magnification. Note pigmented perianal spines which appear dark. Ecdysial scars are also visible.

**Figure 26 insects-16-00088-f026:**
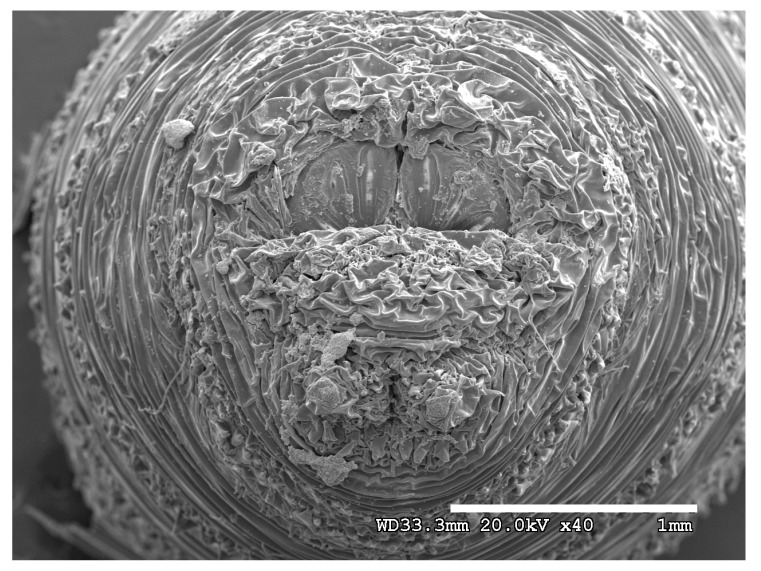
*Cochliomyia hominivorax* puparium. Scanning electron micrograph at 40×, scale bar is 1 mm. Spiracular plates are partially covered by the spiracular labellum.

**Figure 27 insects-16-00088-f027:**
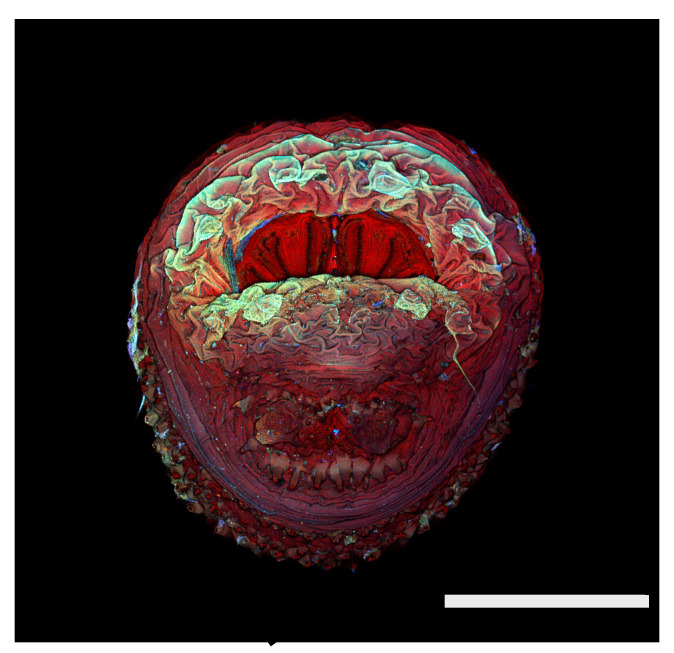
*Cochliomyia hominivorax* puparium. Confocal maximum intensity projection at 40×, scale bar is 1 mm. Spiracular plates are partially covered by the spiracular labellum.

**Figure 28 insects-16-00088-f028:**
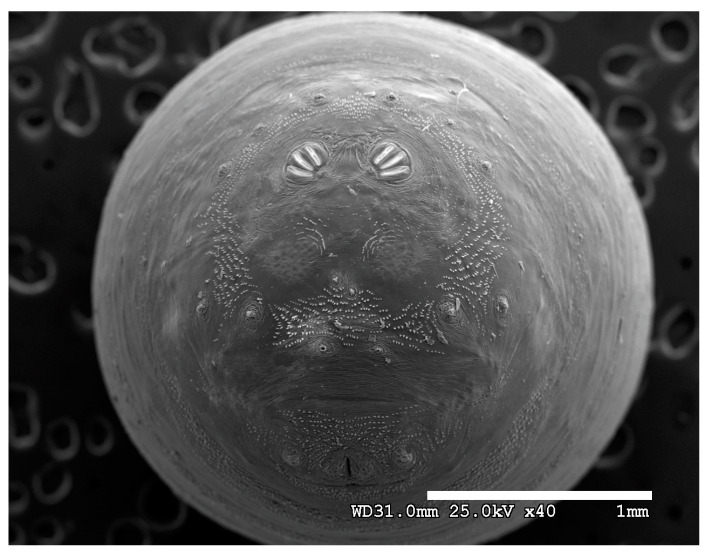
*Lucilia sericata* puparium. Scanning electron micrograph at 40×, scale bar is 1 mm. Spiracular plates with slits and are easily viewed at this minimal magnification, but the spiracular opercula are weak.

**Figure 29 insects-16-00088-f029:**
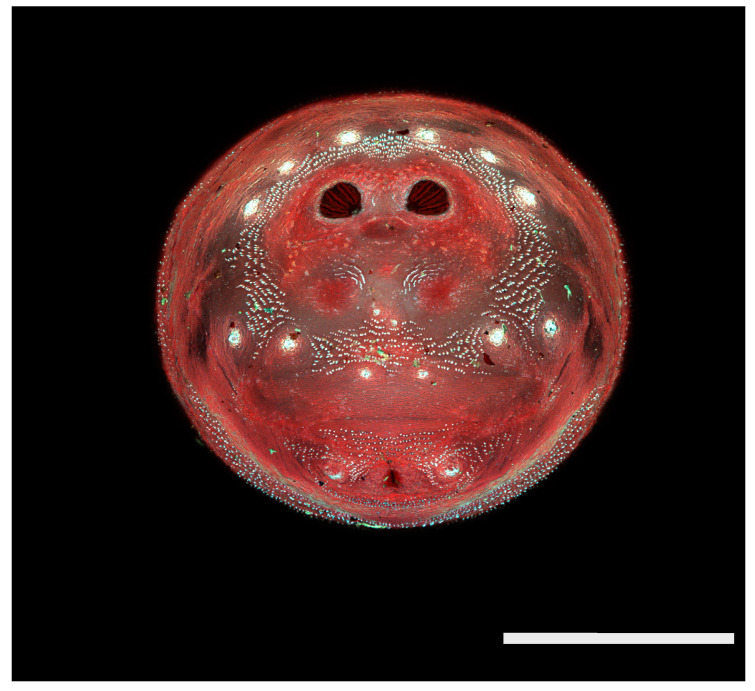
*Lucilia sericata* puparium. Confocal maximum intensity projection at 40×, scale bar is 1 mm. Spiracular plates with slits and weak spiracular opercula are easily viewed at this minimal magnification.

**Figure 30 insects-16-00088-f030:**
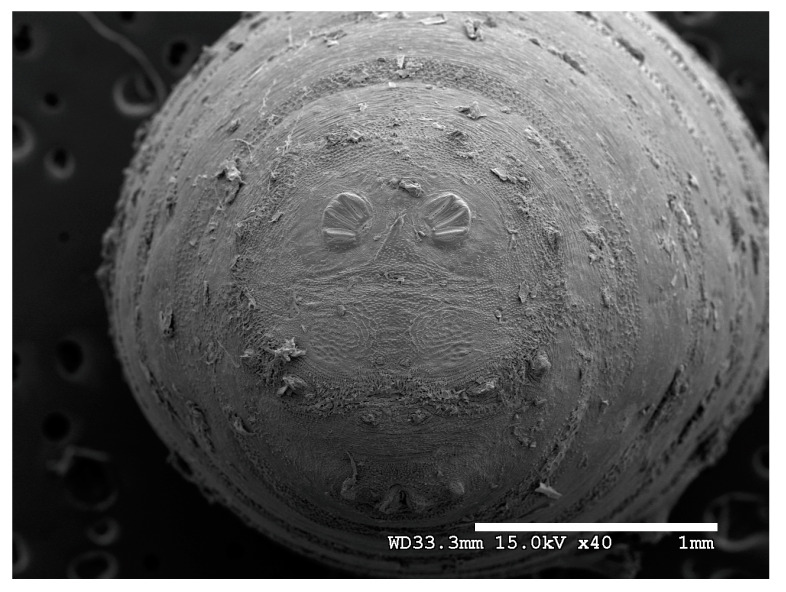
*Lucilia coeruleiviridis* puparium. Scanning electron micrograph at 40×, scale bar is 1 mm. Spiracular plates with slits easily viewed at this minimal magnification. Spiracular opercula difficult to distinguish.

**Figure 31 insects-16-00088-f031:**
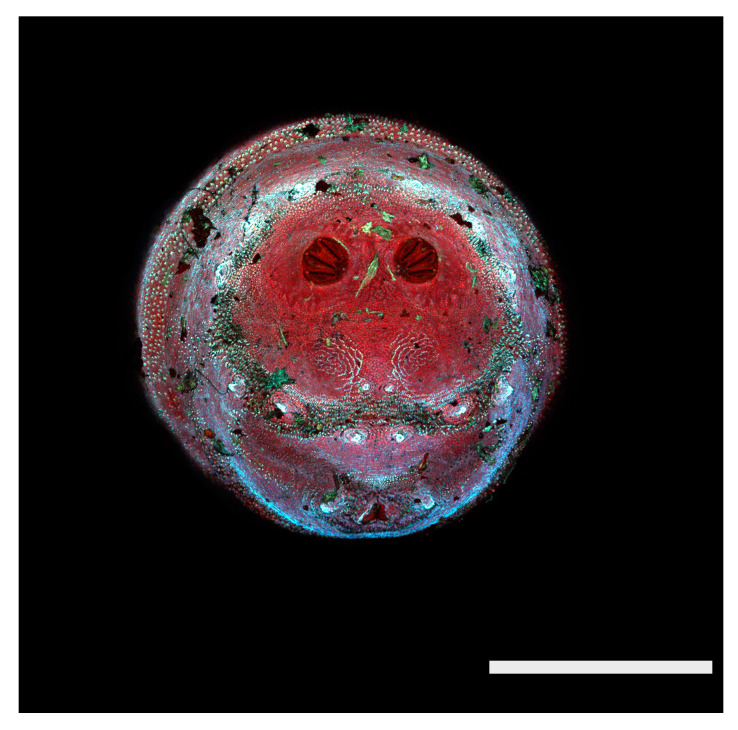
*Lucilia coeruleiviridis* puparium. Confocal maximum intensity projection at 40×, scale bar is 1 mm. Spiracular plates with slits and spiracular opercula are easily viewed at this minimal magnification. While weak, setae of the spiracular opercula are more robust than in *L. sericata*.

**Figure 32 insects-16-00088-f032:**
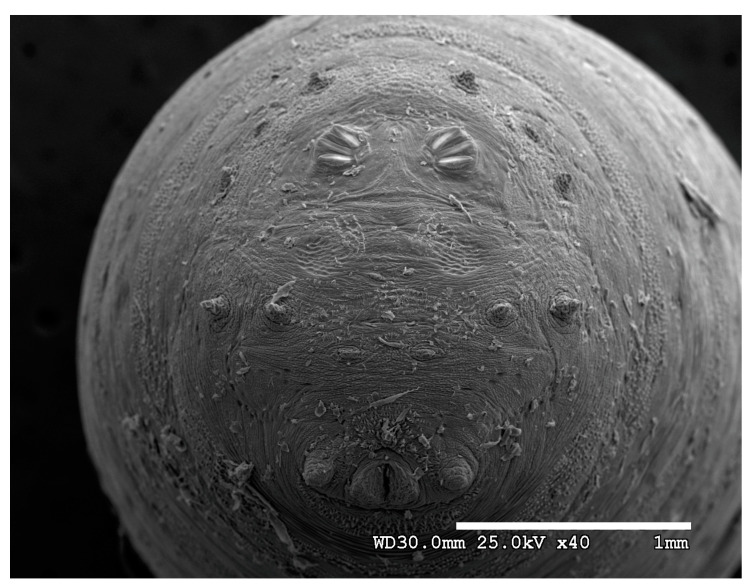
*Cynomya cadaverina* puparium. Scanning electron micrograph at 40×, scale bar is 1 mm. Spiracular plates with slits and spiracular opercula are easily viewed at this minimal magnification. Ecdysial scars are also visible.

**Figure 33 insects-16-00088-f033:**
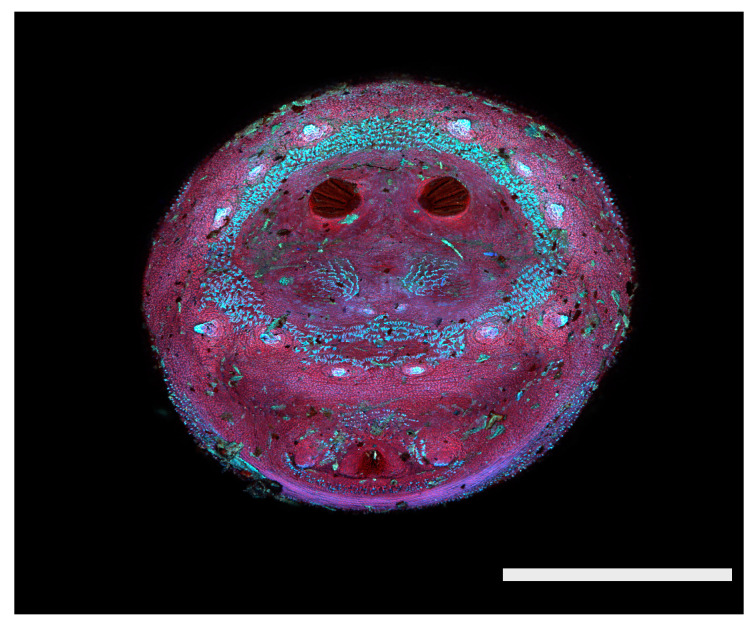
*Cynomya cadaverina* puparium. Confocal maximum intensity projection at 40×, scale bar is 1 mm. Spiracular plates with slits and spiracular opercula are easily viewed at this minimal magnification.

**Figure 34 insects-16-00088-f034:**
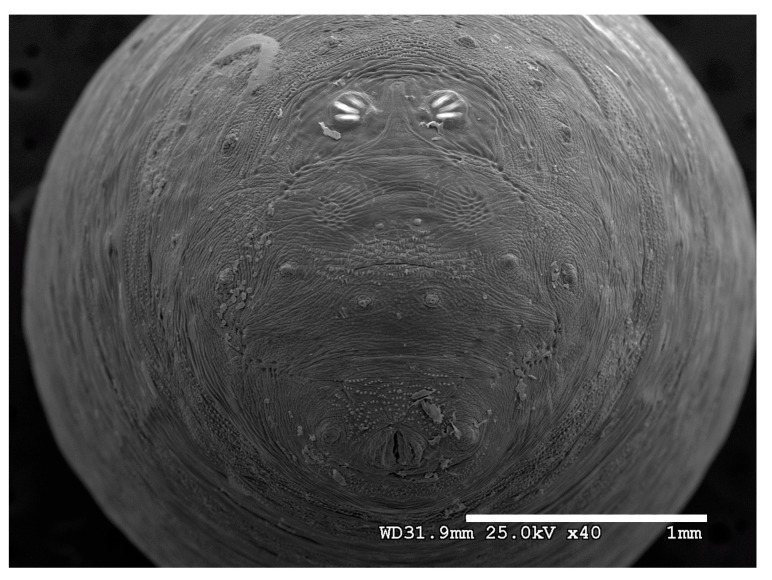
*Calliphora livida* puparium. Scanning electron micrograph at 40×, scale bar is 1 mm. Setae of the spiracular opercula are weak.

**Figure 35 insects-16-00088-f035:**
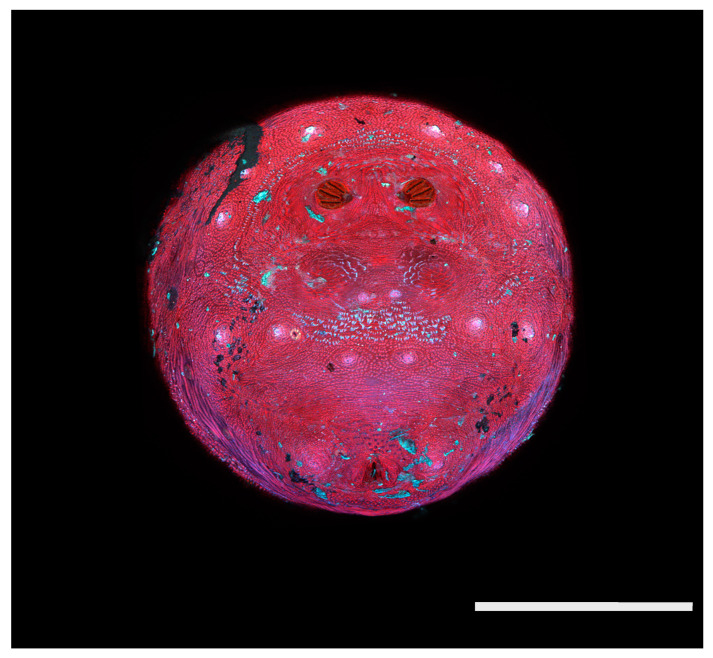
*Calliphora livida* puparium. Confocal maximum intensity projection at 40×, scale bar is 1 mm. Spiracular plates with slits and spiracular opercula are easily viewed at this minimal magnification.

**Figure 36 insects-16-00088-f036:**
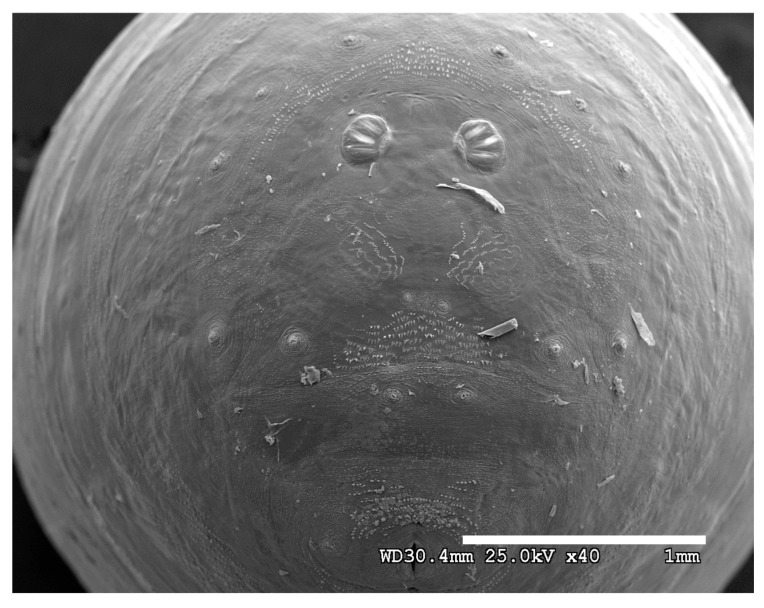
*Calliphora vicina* puparium. Scanning electron micrograph at 40×, scale bar is 1 mm. Spiracular plates with slits and spiracular opercula are easily viewed at this minimal magnification. Ecdysial scars are also visible.

**Figure 37 insects-16-00088-f037:**
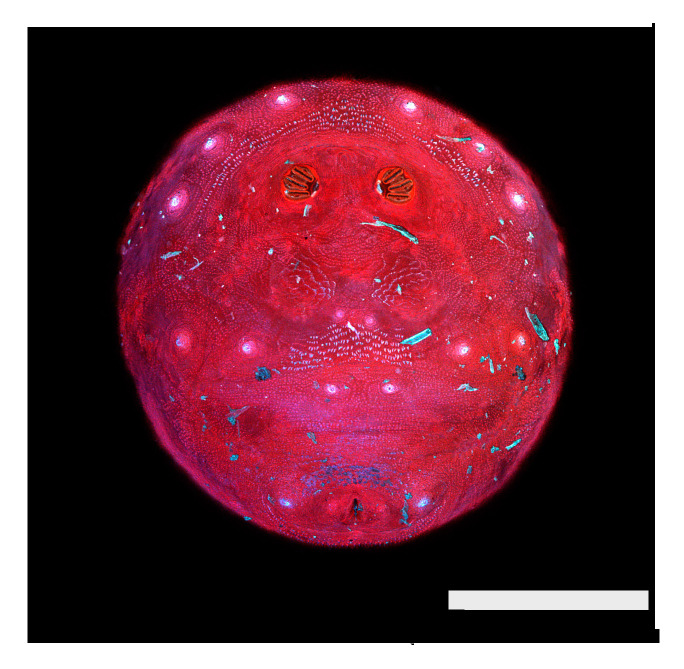
*Calliphora vicina* puparium. Confocal maximum intensity projection at 40×, scale bar is 1 mm. Spiracular plates with slits and spiracular opercula are easily viewed at this minimal magnification.

**Figure 38 insects-16-00088-f038:**
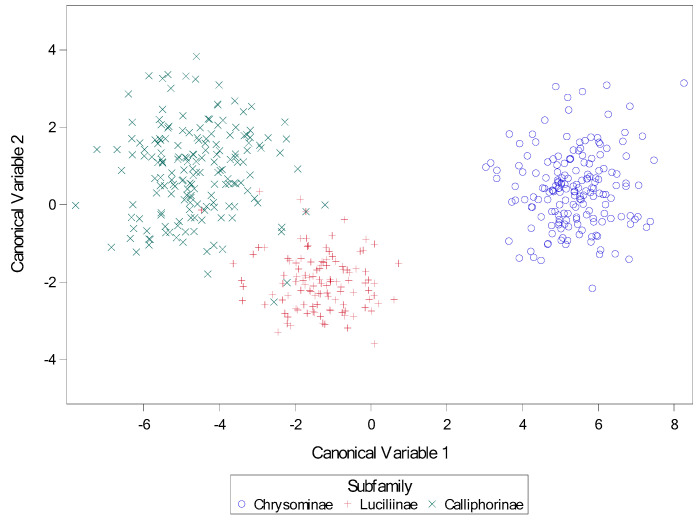
Scatter plot of canonical variates for subfamily. Canonical variables 1 and 2 from canonical discriminate analysis across nine calliphorid species for subfamily. Variables for canonical analysis (based on results from stepwise discriminate analysis) were length2 = the length of the middle slit of the left spiracular plate, angle4 = the angle of the middle slit of the right spiracular plate, and narrow = the narrowest length between plates.

**Figure 39 insects-16-00088-f039:**
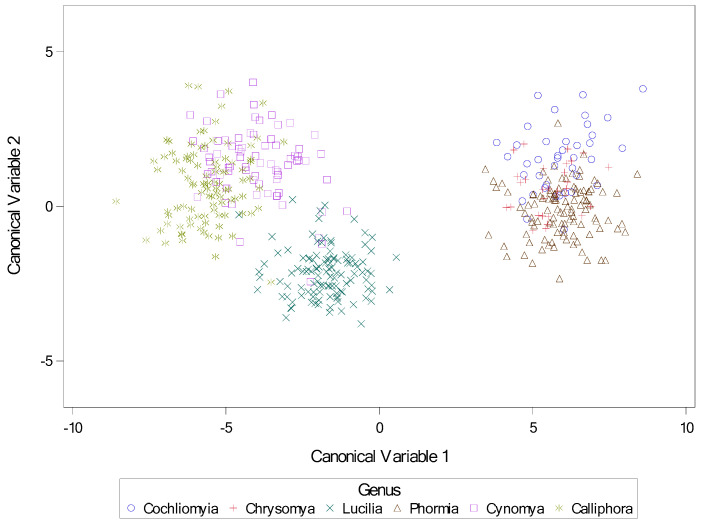
Scatter plot of canonical variables for genus. Canonical variables1 and 2 from canonical discriminate analysis across nine calliphorid species by genus. Variables for canonical analysis (based on results from stepwise discriminate analysis) were length2 = the length of the middle slit of the left spiracular plate, angle4 = the angle of the middle slit of the right spiracular plate, and narrow = the narrowest length between plates.

**Figure 40 insects-16-00088-f040:**
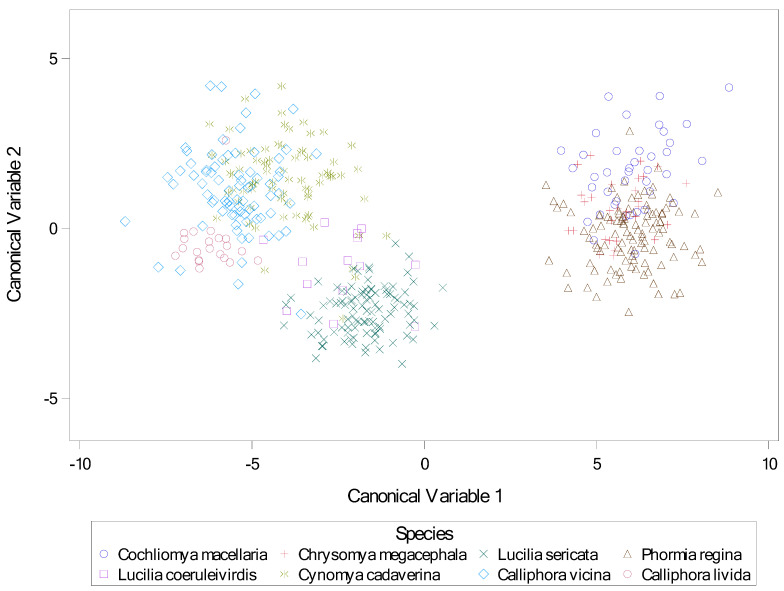
Scatter plot of canonical variables by species. Canonical variables 1 and 2 from canonical discriminate analysis across nine calliphorid species. Variables for canonical analysis (based on results from stepwise discriminate analysis) were length2 = the length of the middle slit of the left spiracular plate, angle4 = the angle of the middle slit of the right spiracular plate, and narrow = the narrowest length between plates.

**Table 1 insects-16-00088-t001:** Background data on specimens. Taxonomic information, number of specimens imaged, location, sources, and diet of calliphorid larvae. A total of 505 individuals were used.

Subfamily	Genus	Species	n	Location	Source	Host/Food
Calliphorinae	*Calliphora*	*livida*	21	Everett, WA, USA	wild	rat
Calliphorinae	*Calliphora*	*vicina*	20	Everett, WA, USA	wild	rat
Calliphorinae	*Calliphora*	*vicina*	57	Everett, WA, USA	wild	rat
Chrysomyinae	*Chrysomya*	*megacephala*	30	Miami, FL, USA	wild	chicken
Chrysomyinae	*Cochliomyia*	*hominivorax*	11	Pacora, Panama ^1^	culture	artificial diet
Chrysomyinae	*Cochliomyia*	*macellaria*	40	Lincoln, NE, USA	wild	pig
Calliphorinae	*Cynomya*	*cadaverina*	76	Lincoln, NE, USA	wild	raccoon
Luciliinae	*Lucilia*	*coeruleiviridis*	14	Lincoln, NE, USA	wild	rabbit
Luciliinae	*Lucilia*	*sericata*	35	Lincoln, NE, USA ^2^	culture	beef liver
Luciliinae	*Lucilia*	*sericata*	64	Lincoln, NE, USA	wild	rat
Chrysominae	*Phormia*	*regina*	39	Lincoln, NE, USA	wild	pig
Chrysominae	*Phormia*	*regina*	28	Everett, WA, USA	wild	rat
Chrysominae	*Phormia*	*regina*	39	Lincoln, NE, USA	culture	beef liver
Chrysominae	*Phormia*	*terraenovae*	31	Bonners Ferry, ID, USA ^3^	culture	fish meal

^1^ Specimens obtained from screwworm production facility under permit from USDA. ^2^ Colony established from wild population in Morgantown, West Virginia. ^3^ Obtained from Grub Co., Fairfield, OH, USA, but produced by Forked Tree Ranch, Inc., Bonners Ferry, ID, USA.

**Table 2 insects-16-00088-t002:** Descriptive statistics for variables used for discrimination by subfamily of calliphorid. Variables are length2 = the length of the middle slit of the left spiracluar plate, angle4 = the angle of the middle slit of the right spiracular plate, and narrow = the narrowest length between plates.

Subfamily		Length2	Angle4	Narrow
Calliphorinae	N	174	174	174
	Mean	42.42563	36.81741	109.8691
	StdErr	0.603716	0.396177	1.018223
Chrysomyinae	N	178	207	207
	Mean	77.45124	65.80676	98.11097
	StdErr	0.359465	0.372499	1.251904
Luciliinae	N	113	113	113
	Mean	45.28814	49.99319	75.27035
	StdErr	0.401046	0.452896	0.868647

**Table 3 insects-16-00088-t003:** Summary results from stepwise canonical discriminate analysis of morphological variables for distinguishing subfamilies of calliphorids (specifically, *Calliphorinae, Chrysominae, and Luciliinae*). Here, a significant F value indicates that the variable cannot be removed from the model (df = 2, 460). Variables are length2 = the length of the middle slit of the left spiracular plate, angle4 = the angle of the middle slit of the right spiracular plate, and narrow = the narrowest length between plates.

Step	Variable	Partial R-Square	F	*p* > F	Wilks’ Lambda	*p* < Lambda	Average Squared Canonical Correlation	*p* > ASCC
1	length2	0.8813	1715.20	<0.0001	0.10439509	<0.0001	0.44780246	<0.0001
2	angle4	0.6692	466.30	<0.0001	0.04189724	<0.0001	0.65483543	<0.0001
3	narrow	0.4617	197.29	<0.0001	0.02771094	<0.0001	0.75597814	<0.0001

**Table 4 insects-16-00088-t004:** Summary univariate test statistics for canonical discriminate analysis of morphological variables for distinguishing subfamilies of calliphorids (specifically, Calliphorinae, Chrysomyinae, and Luciliinae). Variables are length2 = the length of the middle slit of the left spiracular plate, angle4 = the angle of the middle slit of the right spiracular plate, and narrow = the narrowest length between plates.

Variable	Total Standard Deviation	Pooled Standard Deviation	Between Standard Deviation	R-Square	R-Square /(1-RSq)	F_2,462_	*p* > F
length2	17.6095	6.0799	20.2250	0.8813	7.4251	1715.20	<0.0001
angle4	13.9182	5.0221	15.8859	0.8704	6.7138	1550.89	<0.0001
narrow	19.1516	13.7813	16.3078	0.4844	0.9396	217.04	<0.0001

**Table 5 insects-16-00088-t005:** Summary multivariate test statistics and F approximations (S = 0, M = 0, and N = 229) for canonical discriminate analysis of morphological variables for distinguishing subfamilies of calliphorids (specifically Calliphorinae, Chrysomyinae, and Luciliinae).

Statistic	Value	F	Num DF	Den DF	*p* > F
Wilks’ Lambda	0.02113448	901.40	6	920	<0.0001
Pillai’s Trace	1.51598239	481.30	6	922	<0.0001
Hotelling–Lawley Trace	20.90179552	1600.74	6	611.56	<0.0001
Roy’s Greatest Root	19.60551526	3012.71	3	461	<0.0001

**Table 6 insects-16-00088-t006:** Raw canonical coefficients for Can1 and Can2, which were used in separating calliphorid subfamilies of calliphorids (Calliphorinae, Chrysominae, and Luciliinae). Variables are length2 = the length of the middle slit of the left spiracular plate, angle4 = the angle of the middle slit of the right spiracular plate, and narrow = the narrowest length between plates.

Variable	Can1	Can2
length2	0.1342688860	0.0709503297
angle4	0.1677950339	−0.0660280512
narrow	−0.0186688625	0.0627152758

**Table 7 insects-16-00088-t007:** Descriptive statistics for variables used for discrimination by select genera of calliphorids (*Calliphora*, *Chrysomya*, *Cochliomyia*, *Cynomya*, *Lucilia*, and *Phormia*). Variables are length2 = the length of the middle slit of the left spiracular plate, angle4 = the angle of the middle slit of the right spiracular plate, and narrow = the narrowest length between plates.

		Length2	Angle4	Narrow
Calliphora	N	98	98	98
	Mean	37.07776	38.56643	111.8896
	StdErr	0.53704	0.51058	1.326149
Chrysomya	N	30	30	30
	Mean	77.118	65.11933	96.118
	StdErr	0.716633	0.935825	1.731377
Cochliomyia	N	40	40	40
	Mean	76.12175	71.294	115.1433
	StdErr	0.90666	0.639532	1.992577
Cynomya	N	76	76	76
	Mean	49.32158	34.56211	107.2638
	StdErr	0.563951	0.523184	1.543251
Lucilia	N	113	113	113
	Mean	45.28814	49.99319	75.27035
	StdErr	0.401046	0.452896	0.868647
Phormia	N	108	108	108
	Mean	78.0362	65.23676	86.01204
	StdErr	0.439375	0.392917	1.072371

**Table 8 insects-16-00088-t008:** Summary results from stepwise canonical discriminate analysis of morphological variables for distinguishing genera of calliphorids (specifically, *Calliphora*, *Chrysomya*, *Cochliomyia*, *Cynomya*, *Lucilia*, and *Phormia*). Here, a significant F value indicates that the variable cannot be removed from the model (df = 5, 457). Variables are length2 = the length of the middle slit of the left spiracular plate, angle4 = the angle of the middle slit of the right spiracular plate, and narrow = the narrowest length between plates.

Step	Variable	Partial R-Square	F	*p* > F	Wilks’ Lambda	*p* < Lambda	Average Squared Canonical Correlation	*p* > ASCC
1	length2	0.9267	1160.18	<0.0001	0.07332386	<0.0001	0.18533523	<0.0001
2	angle4	0.7059	219.88	<0.0001	0.02156298	<0.0001	0.29147732	<0.0001
3	narrow	0.6135	145.10	<0.0001	0.00833359	<0.0001	0.40665025	<0.0001

**Table 9 insects-16-00088-t009:** Summary univariate test statistics for canonical discriminate analysis of morphological variables for distinguishing genera of calliphorids (specifically, *Calliphora*, *Chrysomya*, *Cochliomyia*, *Cynomya*, *Lucilia,* and *Phormia*). Variables are length2 = the length of the middle slit of the left spiracular plate, angle4 = the angle of the middle slit of the right spiracular plate, and narrow = the narrowest length between plates.

Variable	Total Standard Deviation	Pooled Standard Deviation	Between Standard Deviation	R-Square	R-Square /(1-RSq)	F_5,459_	*p* > F
length2	17.6095	4.7943	18.5496	0.9267	12.6381	1160.18	<0.0001
angle4	13.9182	4.6250	14.3744	0.8908	8.1548	748.61	<0.0001
narrow	19.1516	11.6160	16.7142	0.6361	1.7479	160.46	<0.0001

**Table 10 insects-16-00088-t010:** Summary multivariate test statistics and F approximations (S = 3, M = 0.5, and N = 227.5) for canonical discriminate analysis of morphological variables for distinguishing genera of calliphorids (specifically, *Calliphora*, *Chrysomya*, *Cochliomyia*, *Cynomya*, *Lucilia*, and *Phormia*).

Statistic	Value	F	Num DF	Den DF	*p* > F
Wilks’ Lambda	0.00833359	392.44	15	1262	<0.0001
Pillai’s Trace	2.03325124	193.07	15	1377	<0.0001
Hotelling–Lawley Trace	25.95452800	789.13	15	858	<0.0001
Roy’s Greatest Root	23.40209892	2148.31	5	459	<0.0001

**Table 11 insects-16-00088-t011:** Raw canonical coefficients for Can1 and Can2, which were used in separating genera of calliphorids (specifically, *Calliphora*, *Chrysomya*, *Cochliomyia*, *Cynomya*, *Lucilia*, *I* and *Phormia*). Variables are length2 = the length of the middle slit of the left spiracular plate, angle4 = the angle of the middle slit of the right spiracular plate, and narrow = the narrowest length between plates.

Variable	Can1	Can2
length2	0.1741020171	0.0636367228
angle4	0.1454001990	−0.0571248148
narrow	−0.0208380554	0.0743470086

**Table 12 insects-16-00088-t012:** Descriptive statistics for variables used for discrimination of eight species of calliphorids. Variables are length2 = the length of the middle slit of the left spiracular plate, angle4 = the angle of the middle slit of the right spiracular plate, and narrow = the narrowest length between plates.

Species		Length2	Angle4	Narrow
*Calliphora livida*	N	21	21	21
	Mean	33.41	37.37238	98.51238
	StdErr	0.671428	0.867152	1.663625
*Calliphora vicina*	N	77	77	77
	Mean	38.07805	38.89208	115.5379
	StdErr	0.612128	0.602485	1.35678
*Chrysomya megacephala*	N	30	30	30
	Mean	77.118	65.11933	96.118
	StdErr	0.716633	0.935825	1.731377
*Cochliomya macellaria*	N	40	40	40
	Mean	76.12175	71.294	115.1433
	StdErr	0.90666	0.639532	1.992577
*Cynomya cadaverina*	N	76	76	76
	Mean	49.32158	34.56211	107.2638
	StdErr	0.563951	0.523184	1.543251
*Lucilia coeruleivirdis*	N	14	14	14
	Mean	46.20071	46.13857	85.63643
	StdErr	1.409753	1.914886	3.311618
*Lucilia sericata*	N	99	99	99
	Mean	45.15909	50.53828	73.80444
	StdErr	0.413709	0.417504	0.775613
*Phormia regina*	N	108	108	108
	Mean	78.0362	65.23676	86.01204
	StdErr	0.439375	0.392917	1.072371

**Table 13 insects-16-00088-t013:** Summary results from stepwise canonical discriminate analysis of morphological variables for distinguishing eight species of calliphorids (see text for list). Here, a significant F value indicates that the variable cannot be removed from the model (df = 7, 455). Variables are length2 = the length of the middle slit of the left spiracular plate, angle4 = the angle of the middle slit of the right spiracular plate, and narrow = the narrowest length between plates.

Step	Entered	Partial R-Square	F	*p* > F	Wilks’ Lambda	*p* < Lambda	Average Squared Canonical Correlation	*p* > ASCC
1	length2	0.9293	857.71	<0.0001	0.07073250	<0.0001	0.13275250	<0.0001
2	angle4	0.7149	163.32	<0.0001	0.02016849	<0.0001	0.21007083	<0.0001
3	narrow	0.6494	120.39	<0.0001	0.00707114	<0.0001	0.29735764	<0.0001

**Table 14 insects-16-00088-t014:** Summary univariate test statistics for canonical discriminate analysis of morphological variables for distinguishing eight species of calliphorids (see text for list). Variables are length2 = the length of the middle slit of the left spiracular plate, angle4 = the angle of the middle slit of the right spiracular plate, and narrow = the narrowest length between plates.

Variable	Total Standard Deviation	Pooled Standard Deviation	Between Standard Deviation	R-Square	R-Square /(1-RSq)	F_7,457_	*p* > F
length2	17.6095	4.7191	18.1278	0.9293	13.1378	857.71	<0.0001
angle4	13.9182	4.5696	14.0520	0.8938	8.4191	549.65	<0.0001
narrow	19.1516	11.0136	16.7940	0.6743	2.0701	135.15	<0.0001

**Table 15 insects-16-00088-t015:** Summary multivariate test statistics and F approximations (S = 3, M = 1.5, and N = 226.5) for canonical discriminate analysis of morphological variables for distinguishing nine species of calliphorids (see text for list).

Statistic	Value	F	Num DF	Den DF	*p* > F
Wilks’ Lambda	0.00707114	286.90	21	1307.1	<0.0001
Pillai’s Trace	2.08150350	147.95	21	1371	<0.0001
Hotelling–Lawley Trace	27.10464780	585.93	21	944.74	<0.0001
Roy’s Greatest Root	24.16657519	1577.73	7	457	<0.0001

**Table 16 insects-16-00088-t016:** Raw canonical coefficients for Can1 and Can2, which were used in separating eight calliphorid species (see text for species list). Variables are length2 = the length of the middle slit of the left spiracular plate, angle4 = the angle of the middle slit of the right spiracular plate, and narrow = the narrowest length between plates.

Variable	Can1	Can2
length2	0.1756548037	0.0635858969
angle4	0.1490606592	−0.0557843066
narrow	−0.0190332814	0.0803654761

## Data Availability

Data is contained within the article or Appendix A.

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
