# Peer review of "Confocal Laser Scanning Microscopy as a Method for Identifying Variation in Puparial Morphology and Establishing Characters for Taxonomic Determination"

_insects, 2025, doi:10.3390/insects16010088_

Round 1

Reviewer 1 Report

Comments and Suggestions for Authors

This article addresses a method for identifying variation in puparial morphology and establishing characters for taxonomic determination in blowflies of forensic interest. Several aspects need to be highlighted:

- on line 16: laser...laser?

- line 48: larvae sizes: factor little explored in the text;

- line 52: calliphorids;

- line 86: complicated (why?) combinations of...

- line 129: adult emergence: because of the larvae eclosion;

- line 153: statistical analysis: which?

- line 186: would Figures 1 and 2 show changes depending on the size of the puparium?

- lines 247 and 678: C. macellaria in italics;

- line 292: points of/with? narrowest...

- lines 295-296: to indicate ... was indicated...?

- lines 399 and 1218: "perispiracular papillae" and "dentate setae" not indicated in the Figure;

- line 406: only three...(indicated much later in the lines 1050-1053);

- line 427: Figure 16 quoted immediately after the Figure 9;

- lines 583 and 598: (Figure 10) and (Figure 16), respectively;

- line 674: Figures;

I think that there was a lack of a more detailed interpretation showing possible limitations due to overlaps between genera and species in the Figures 39 and 40;

- lines 1294-1296: new data that don´t appear before in the results?

- References: very few were cited. There are others about pupal development; and the article of Voss et al. (2017) (Int. J. Leg. Med.)?

Author Response

This article addresses a method for identifying variation in puparial morphology and establishing characters for taxonomic determination in blowflies of forensic interest. Several aspects need to be highlighted:

  • on line 16: laser...laser? \
  • fixed
  •  
  • line 48: larvae sizes: factor little explored in the text;
  • we think this is adequately addressed in the reference cited.
  •  
  • line 52: calliphorids;
  • fixed
  •  
  • line 86: complicated (why?) combinations of...
  • fixed
  • line 129: adult emergence: because of the larvae eclosion;
  • fixed
  • line 153: statistical analysis: which?
  • fixed
  • line 186: would Figures 1 and 2 show changes depending on the size of the puparium?
  • locations and relationships would be consistent across sizes
  • lines 247 and 678: C. macellaria in italics;
  • fixed
  • line 292: points of/with? narrowest...
  • fixed
  • lines 295-296: to indicate ... was indicated...?
  • fixed
  • lines 399 and 1218: "perispiracular papillae" and "dentate setae" not indicated in the Figure;
  • fixed in figure legends
  • line 406: only three...(indicated much later in the lines 1050-1053);
  • fixed
  • line 427: Figure 16 quoted immediately after the Figure 9;
  • fixed
  • lines 583 and 598: (Figure 10) and (Figure 16), respectively;
  • fixed
  • line 674: Figures;
  • fixed

I think that there was a lack of a more detailed interpretation showing possible limitations due to overlaps between genera and species in the Figures 39 and 40;

added text for explanation

  • lines 1294-1296: new data that don´t appear before in the results?
  • no, this just follows from the results presented
  • References: very few were cited. There are others about pupal development; and the article of Voss et al. (2017) (Int. J. Leg. Med.)?
  • added reference.

Reviewer 2 Report

Comments and Suggestions for Authors

Dear AA, I read your manuscript with interest because as an entomologist I have done numerous works on Diptera using mainly SEM. The work is well done even if the introduction needs some adjustments and seems too long to me. There is no discussion about the quality of the images obtained with the confocal microscope, but I have serious doubts that this instrument is a complete substitute for both traditional and electronic microscopy. Some images made with the SEM microscope are, in my opinion, superior to those obtained with the confocal microscope which also requires a considerable basic preparation of the operator. However, I renew my compliments for the work that was conducted with scientific rigor also considering the statistical part that reinforces the use of this "new" means of observation. I have noted a few things that I attach in a separate file.

Author Response

Line 41: Erase literally

fixed

Line 52: Calliphorids and to produce…..

fixed
Lines 70-71: this sentence is not clear for the common reader, please specify better

fixed 
Lines 74-79: again, it is a sentence directed to specialists in the microscopy sector and not to readers, I
recommend removing it

upon reflection, we decided to retain this sentence as it seems likely microscopy specialists would be readers of the paper
Lines 1206-1214: I disagree with this sentence, since there are now a variety of SEM’s that are able to allow
a direct observation until 10-14 samples/session, working at low vacuum and not requiring any complicate
procedure in a very faster way.

While the reviewer is correct that some faster methods exist, we disagree that our characterization of general limitations with SEM is incorrect. Consequently we retained this text

Lines 216-217: this speculation is not appropriate and I think not right, please remove the sentence.

fixed 

Line 1254: remove citation of tabs.

fixed

Reviewer 3 Report

Comments and Suggestions for Authors

In the reviewed MS, two researchers from the U.S. report on their results of comparative morphological studies of puparia of calliphorid flies. They investigated a set of morphometrics and determined statistically a group of characters which can be used of discriminating different taxa of calliphorids. The statistical analyses were carried out correctly. The MS includes a lot of high-quality images, however, they are too large and numerous. The authors could consider combining several images into one plate, this would help to reduce the number of figures. The Sections Results and Discussion need reconsideration and restructuring. They are written in suboptimal way. As a result, it is hard to capture the main idea of each paragraph in these sections. The authors could structure Results and Discussion in correspondence with their goals 1-4 mentioned in the section 1.2. In general, this MS resembles a chapter from a PhD thesis. It would be better to rearrange it stylistically and transform it to a more condensed, shorter scientific text, which is more typical for scientific papers. Some additional remarks are below.

33 - Cyclorrhapous - please check the taxon name

51-53 would it be more useful to perform classical morphological study of the puparia using common microscopy prior to using CLSM, which is notably rarer in comparison to common light microscopy?

62 - bombards – please consider using another verb

94 - the same author published a series of paper on CLSM for mites. This citation seems superficial. Please, consider reviewing papers by Spanish author Valdecasas who published several interesting papers on CLSM for mites. Additionally, CLSM was applied for examination of ancient amber inclusions by palaeoentomologists, this aspect could also be mentioned in this section.

135 - please, indicate how many hours/days it took to image the 509 specimens.

160 - CGE, LGH, and Neal Haskell – please, explain who are these guys? Are they the authors or collaborators with the same affiliation?

Section 2.1 Did you perform comparison of the CLSM signal of fresh and ethanol fixed material?

180 - After initial evaluation of character states (essentially, identifying characters that seemed to show intraspecific stability but interspecific variability) --- please explain this better: how exactly was it done? Which characters were excluded and which were included in further analysis

183 - 12 points were identified by the user – please, provide a better explanation of the point choice. Please, give some introductory information on the slits (e.g in a short section in Introduction), otherwise all these may be unclear for a reader

Please, consider combining images 1,2,3 into a single plate. No need to give complete image for all species, it could be done for only 1 of them because the characters of interested are located in a small central area.

Fig2a Cochliomyia macellaria – italic

Fig2a – please, explain how the measurements of angles was standardized for all cases? The green line needs a better explanation.

Fig2b Dotted grey line – it looks white to me because the background is gray

Fig2b what are the black triangles?

Section 2.3 – did you perform standardization of the mmeasurementsor not? It is not clear from the text

423-428 – could you give a subheading for this paragraph? The main idea of this paragraph is unclear? Which result exactly do you want to report here?

427 – When you say on SEM, please give some distinct comparative data (CLSM vs SEM) otherwise is unclear how did you come to these conclusions. You may give a Table and compare different characters for both methods.

435-445 This paragraph suffers the same problem: it is unclear which result you want to present here. Please, reconsider the text.

1048 you could start a new subsection of results here.

Statistical results: please provide a distinct conclusion – which characters are appropriate for species/genus/subfamily identification?

1206-1247 please, consider giving comparative data (CLSM vs SEM) in Results and your interpretation - in the Discussion

You could structure the Results and the Discussion in correspondence with the goals 1-4 listed in the section 1.2. In this case Results would have 4 paragraphs (each reporting the results related with corresponding goal) and Discussion would also consist of 4 paragraphs, each of them could have a subheading corresponding to one of the goal 1-4.

Author Response

In the reviewed MS, two researchers from the U.S. report on their results of comparative morphological studies of puparia of calliphorid flies. They investigated a set of morphometrics and determined statistically a group of characters which can be used of discriminating different taxa of calliphorids. The statistical analyses were carried out correctly. The MS includes a lot of high-quality images, however, they are too large and numerous. The authors could consider combining several images into one plate, this would help to reduce the number of figures. The Sections Results and Discussion need reconsideration and restructuring. They are written in suboptimal way. As a result, it is hard to capture the main idea of each paragraph in these sections. The authors could structure Results and Discussion in correspondence with their goals 1-4 mentioned in the section 1.2. In general, this MS resembles a chapter from a PhD thesis. It would be better to rearrange it stylistically and transform it to a more condensed, shorter scientific text, which is more typical for scientific papers. Some additional remarks are below.

33 - Cyclorrhapous - please check the taxon name

ok as written

51-53 would it be more useful to perform classical morphological study of the puparia using common microscopy prior to using CLSM, which is notably rarer in comparison to common light microscopy?

Light microscopy is likely to remain the most common method for studying puparial morphology, but the advantages of CLSM, as we try to show here, are the much higher resolution and the fluorescence of cuticular waxes associated with specific character states. Objective three of in the introduction highlights the potential usefulness of CLSM images in guiding work with conventional light microscopy

62 - bombards – please consider using another verb

fixed

94 - the same author published a series of paper on CLSM for mites. This citation seems superficial. Please, consider reviewing papers by Spanish author Valdecasas who published several interesting papers on CLSM for mites. Additionally, CLSM was applied for examination of ancient amber inclusions by palaeoentomologists, this aspect could also be mentioned in this section.

citations added -- many thanks

135 - please, indicate how many hours/days it took to image the 509 specimens.

ca. 15 min. per individual - added

160 - CGE, LGH, and Neal Haskell – please, explain who are these guys? Are they the authors or collaborators with the same affiliation?

affliliations added

Section 2.1 Did you perform comparison of the CLSM signal of fresh and ethanol fixed material?

Visually yes, and we observed no obvious differences. Statistically, we didn't do a comparison.

180 - After initial evaluation of character states (essentially, identifying characters that seemed to show intraspecific stability but interspecific variability) --- please explain this better: how exactly was it done? Which characters were excluded and which were included in further analysis

text revised to clarify determination of character states

183 - 12 points were identified by the user – please, provide a better explanation of the point choice. Please, give some introductory information on the slits (e.g in a short section in Introduction), otherwise all these may be unclear for a reader

revised text to make obvious the 12 points are associated with the six spiracular slits. 

Please, consider combining images 1,2,3 into a single plate. No need to give complete image for all species, it could be done for only 1 of them because the characters of interested are located in a small central area.

We retained the current figure structure to allow for showing the characters in the context of the entire posterior of the puparium. Also, we think keeping the figures as is provides better resolution of the key characters than would be possible if combined into a single plate.

Fig2a Cochliomyia macellaria – italic

fixed

Fig2a – please, explain how the measurements of angles was standardized for all cases? The green line needs a better explanation.

The measurement of angles was conducted through the provided macro developed for the ImageJ software. This is indicated in lines 184-186.

Fig2b Dotted grey line – it looks white to me because the background is gray

A comparison of the dotted line with the white background shows the dotted line is light gray.

Fig2b what are the black triangles?

These are papillae, which are indicated in the figure legend

Section 2.3 – did you perform standardization of the mmeasurementsor not? It is not clear from the text

As indicated in this section all measurement were made from the ImageJ macro, so they are not standardized in any way.

423-428 – could you give a subheading for this paragraph? The main idea of this paragraph is unclear? Which result exactly do you want to report here?

revised 

427 – When you say on SEM, please give some distinct comparative data (CLSM vs SEM) otherwise is unclear how did you come to these conclusions. You may give a Table and compare different characters for both methods.

The sentence indicates the deficiencies of SEM, namely being destructive, expensive, and time consuming. As these are the key points for differences between SEM and CLSM, we don't think another table is needed.

435-445 This paragraph suffers the same problem: it is unclear which result you want to present here. Please, reconsider the text.

revised, topic sentence added

1048 you could start a new subsection of results here.

done

Statistical results: please provide a distinct conclusion – which characters are appropriate for species/genus/subfamily identification?

The key characters --  length2 = the length of the middle slit of the left spiracular plate, angle4 = the angle of the middle slit of the right spiracular plate, and narrow = the narrowest length between plates -- are highlighted multiple times in the text of the table headings as well as in the tables themselves.

1206-1247 please, consider giving comparative data (CLSM vs SEM) in Results and your interpretation - in the Discussion

we have offered considerable discuss on SEM limitations and advantages associated with CLSM in the discussion

You could structure the Results and the Discussion in correspondence with the goals 1-4 listed in the section 1.2. In this case Results would have 4 paragraphs (each reporting the results related with corresponding goal) and Discussion would also consist of 4 paragraphs, each of them could have a subheading corresponding to one of the goal 1-4.

We considered this change in organization but decided we would loose too much information that is in the current structure of the results in discussion. We did add subheadings to sections of the results that should clarify the structure and purpose of each section.

Round 2

Reviewer 3 Report

Comments and Suggestions for Authors

The authors carefully revised the MS according to the remarks by reviewer. I have no major criticism on this MS. Please, check the typos in the line 1526: Eriophyoidea